# Growth Rate Prediction, Performance, and Biochemical Enhancement of Black Soldier Fly (*Hermetia illucens*) Fed with Marine By-Products and Co-Products: A Potential Value-Added Resource for Marine Aquafeeds

**DOI:** 10.3390/insects16020113

**Published:** 2025-01-23

**Authors:** Daniela P. Rodrigues, Ricardo Calado, Marisa Pinho, M. do Rosário Domingues, José Antonio Vázquez, Olga M. C. C. Ameixa

**Affiliations:** 1ECOMARE, CESAM—Centre for Environmental and Marine Studies, Department of Biology, University of Aveiro, Santiago University Campus, 3810-193 Aveiro, Portugal; rjcalado@ua.pt (R.C.); olga.ameixa@ua.pt (O.M.C.C.A.); 2CESAM—Centre for Environmental and Marine Studies, Department of Chemistry, University of Aveiro, Santiago University Campus, 3810-193 Aveiro, Portugal; marisapinho@ua.pt (M.P.); mrd@ua.pt (M.d.R.D.); 3Mass Spectrometry Centre, LAQV REQUIMTE, Department of Chemistry, University of Aveiro, Santiago University Campus, 3810-193 Aveiro, Portugal; 4Group of Recycling and Valorisation of Waste Materials (REVAL), Marine Research Institute (IIM-CSIC), C/Eduardo Cabello, 6, CP 36208 Vigo, Galicia, Spain

**Keywords:** insect feed, PUFA, circular bioeconomy, fish by-product management, fish protein hydrolysates

## Abstract

Aquafeed production is a rapidly growing industry focused on sustainable practices, seeking cost-efficient raw materials to enhance its ingredient portfolio, traditionally dominated by fish meal and oil. Insect meals, particularly those produced with black soldier fly (BSF) larvae, have demonstrated potential as a sustainable alternative. In this study, by-products from tuna heads (THs) and codfish frames (CFs), hydrolysates of TH and CF, and TH oils were fed in different incorporation levels to BSF larvae to assess improvements in their nutritional profile, particularly in omega-3 essential fatty acid and protein content. The results revealed that high levels of hydrolysed by-products negatively affected BSF larval survival, while substrate characteristics, such as moisture and protein content, influenced bioconversion rates. Despite this, BSF larvae fed with these by-products exhibited valuable amino acid (lysine 5.8–8.4%, methionine 1.5–2.4%) and omega-3 fatty acid (13.8%) profiles, supporting the inclusion of BSF meal in aquafeeds. These findings highlight BSF’s potential to upcycle fish industry by-products, contributing to more sustainable aquafeed production within a circular bioeconomy framework.

## 1. Introduction

An urge to find additional sources of nutrients in a world with an increasing population, where natural resources are being overexploited, has led to new trends in human and animal nutrition [1]. Aquaculture is seen as a strong candidate in the search for healthy protein and lipid sources, and that area is expected to grow in the upcoming years [2,3]. However, several questions are raised due to the amount of feed needed to supply the animal species commonly produced in this industry, namely because a significant part of the ingredients used in aquafeeds still promote environmental constraints [4]. Fish meal and fish oil, soybeans, oilseeds, vegetable meals, etc., are some of the ingredients most commonly employed in the formulation of aquafeeds [5,6]. The environmental footprint of each ingredient is a reality that cannot be neglected anymore and has gained much interest in recent years [7,8,9]. Moreover, fish meal and fish oil from fresh fish have declined in production due to strategic limitations on the capture of pelagic fish to maintain natural fish stocks [3]. In this scenario, the search to diversify the sources of protein and lipids with a low ecological footprint has gained momentum, namely for the formulation of aquafeeds [8,9,10]. As such, several by-products from the food industry (particularly the fish industry) and agriculture have gained relevance in the last few years [11,12]. Portugal and Spain are two Southern European countries leading the production of canned seafood and cod salting [13,14,15].

The side streams of these activities can range from wastewaters (brine) to blood, viscera, scales, heads and tails, or fish bones (frames) [16]. These by-products are rich in proteins and lipids, particularly free amino acids and highly unsaturated fatty acids (HUFAs), that have a high nutritional value and are associated with important biological and physiological functions in animals and humans [17,18,19]. Their inclusion in feeds and foods represents a strategic waste reduction approach, allied with the closure of the nutrient cycle [20,21]. In this regard, insects are an evident option to pursue these goals, as they showcase the ability to thrive in a wide range of organic substrates needing no prior treatment and generating an exceptional volume of biomass with a nutritional profile able to fulfil the nourishment needs of several species (including humans) [22,23,24]. Black soldier fly larvae (BSFLs) have received great attention due to their ability to convert by-products into high-value biomass and the capacity to modulate their nutritional profile with the rearing substrate [25,26,27,28,29]. For that purpose, a considerable amount of research has been developed to include HUFAs in the lipidic profile of BSFLs destined to be used as fish feed [27,30,31,32].

However, the levels of inclusion of HUFAs in BSFLs are far from the needs of most cultured marine species that must be supplied with aquafeeds, and some authors have proposed that these larvae display a limited ability for the inclusion of certain HUFAs, such as eicosapentaenoic acid (EPA) and docosahexaenoic acid (DHA) [33]. Yet, to the best of the authors’ knowledge, an optimisation of the bioavailability of these biomolecules in substrates of marine origin to produce BSFLs has never been attempted. Hydrolysis is a well-recognised biotechnological procedure mediated by enzymes that displays reduced costs and requires the use of only a few chemicals and soft operation conditions to be efficiently employed [34,35]. The resulting hydrolysates are rich in free amino acids, peptides, and hydrosoluble proteins [36]. Additionally, fat molecules are also released from their initial structures and remain freely in the hydrolysate, making them more accessible to be assimilated or for their efficient recovery by centrifugation.

In the present work, the by-products and co-products from two important fish processing industries in the Iberian Peninsula, tuna heads (THs) and codfish frames (CFs) (produced by the canning and the codfish-salting industries, respectively), were supplied in different levels of replacement of a control diet to BSFLs, either untreated or hydrolysed, in order to understand if it is possible to improve the incorporation of EPA and DHA, as well as enhance the protein/amino acid content in BSFLs fed with these substrates. Moreover, a proof-of-concept trial was also performed with oil retrieved from TH hydrolysis and supplied to a BSF larvae diet, aiming to enhance the incorporation of *n*-3 fatty acids (FAs) and the general larval performance of BSF.

## 2. Materials and Methods

### 2.1. Experimental Diets

#### 2.1.1. Tuna Head (TH) and Atlantic Codfish Frame (CF) By-Products 

Tuna heads were obtained as by-products from yellowfin tuna (*Thunnus albacares*) industrial processing and kindly supplied by Valora Marine Ingredients (Jealsa Corporation, Boiro, A Coruña, Spain). Frames from Atlantic codfish (*Gadus morhua*) were obtained from an industrial cod-salting unit (Bacalhau do Barents—Produtos Alimentares Lda, Gafanha da Nazaré, Portugal). All biological material was minced, up to a size of 20–40 × 20–40 mm, using a meat mincer (Mobba PM32, Mobba Industrial Catalunya, Badalona, Spain) and stored at −18 °C until further use. These raw materials were then freeze-dried (LyoBeta, Syntegon Telstar SLU, Terrassa, Spain) to obtain crude substrates that were studied as potential feed ingredients (THs and CFs) for BSF larvae. The amount of processed materials was around 20 kg of each by-product.

#### 2.1.2. Production of Fish Protein Hydrolysates (FPHs) from Tuna Heads and Codfish Frames 

Lab-scale hydrolysis was performed in a 5 L glass reactor (pH-Stat system equipped with additional temperature, agitation, and reagents addition control), mixing 2 kg of THs or CFs in 2 L of distilled water (solid/liquid ratio of (1:1)), employing 5 mol/L NaOH for pH control. The experimental conditions of the hydrolysis were established on the optimal values indicated for each substrate: (i) 56.8 °C/pH 8.35/[alcalase] = 0.25% (*v*/*w*)/3 h of hydrolysis for cod frames [37], and (ii) 58.5 °C/pH 8/[alcalase] = 0.50% (*v*/*w*)/3 h of hydrolysis for tuna heads [38]. Alcalase 2.4 L (2.4 Anson Unit/g, AU/g enzyme) was the protease utilised for the digestion of industrial by-products and was purchased from Novozymes (Nordisk, Bagsvaerd, Denmark). At the end of the enzymatic digestion process (3 h), bones were removed by filtration (100 μm). The oils from the THs were recovered by centrifugation (15,000× *g* for 20 min) and decantation (for 15 min) from liquid hydrolysate. For tuna, another similar hydrolysis procedure was performed, using the same conditions of operation described above, in which oils were not separated by centrifugation from liquid FPH. Both hydrolysates from cod and from tuna (in this case including oils) were fast-warmed (90 °C for 15 min) for alcalase inactivation (Figure 1) and lyophilised until further use (THH and CFH). The TH oil (THO) recovered during processing was stored in an amber bottle at −20 °C, with nitrogen being injected into the upper part of the recipient to prevent lipid oxidation. A scheme detailing the production of the different fish substrates employed in the present work is presented in Figure 1.

### 2.2. Black Soldier Fly Rearing

All BSFLs employed in the present study were collected from a previously reared generation of specimens originating from our rearing facilities at ECOMARE, University of Aveiro (Portugal). As referenced in other studies [7,31,39,40], a control diet consisting of chick feed (Zêzere A-104 Pintos Migalha: crude protein 17%, crude lipid 3.5%, ash 5%, and crude fibre 5.5%) and tap water (1:1, *v*/*v*) was fed to the BSFLs, with these being kept under controlled conditions (a photoperiod of 16L:8D, at 40 ± 5% of relative humidity and an average room temperature of 30 ± 3 °C). All larvae used in the tests performed in the present study were selected 7 days post-hatching. To ensure optimal larval development up to the feeding trials, newly hatched larvae were fed on the control diet (chicken feed) until they were harvested. Each group, with 50 individuals, was then weighed on an electronic weighting scale (using a RADWAG AS 220.R2 analytical balance) to determine its collective average weight and consequently the mean initial weight per group. The larvae were then transferred to cylindrical plastic containers of 120 mL with ventilated caps. Each container was filled with the corresponding testing substrate at a rate of 15 mg/larva/day. Each substrate treatment was tested using seven independent replicates.

### 2.3. Feeding Trials

Seven-day fed larvae were supplied with the pre-treated raw materials THs and CFs, with the following levels of replacement of the control diet (chicken feed): 100%, 50%, and 25%. These treatments were prepared as follows: (1) lyophilisation of ground raw material (TH100, TH50, TH25, CF100, CF50, CF25); (2) enzymatic hydrolysis with subsequent lyophilisation (THH100, THH50, THH25, CFH100, CFH50, CFH25); and (3) lipid fraction collection from the enzymatic hydrolysis of THs (THO), later incorporated into the control diet in percentages of 10%, 5%, and 2.5% (*w*/*w*) (THO10, THO5, THO2.5). Codfish frames displayed low levels of crude fat (<5%); thus, the lipid fraction was not recovered after enzymatic hydrolysis (Figure 1).

Weight gain was controlled by weighing a random sample of 10 larvae during the trials (using a RADWAG AS 220.R2 analytical balance) and returning them to their container of origin again. On the 10th day (the 17th day of the life cycle), the larvae were harvested, already at the pre-pupal stage (when larvae reduce their feeding activity and begin to develop a darker body pigmentation and initiate their migration from the feeding substrate; this pupal stage is the most used BSF life stage in insect meal production); they were weighed, and the survival rate was determined. After this, they were starved for 24 h (to secure gut cleaning). Lastly, they were freeze-dried for further analysis of their lipid fraction, as well as their ash, protein, fatty acids (FAs), and amino acids (AAs). All feeding experiments were performed under the same BSF feedstock-controlled conditions (namely of temperature, photoperiod, and humidity).

### 2.4. Larval Performance and Conversion Efficiency

Survival (*S*) was calculated using the following equation:(1)$$S=\left(\frac{L_{End}}{L_{Initial}}\right)\times 100$$
where *L_End_* represents the number of larvae at the end of the trial and *L_Initial_*, the initial number of larvae.

Bioconversion efficiency (*BCE*) was calculated as follows:(2)$$BCE=(\frac{\left({ML}_{End}-{ML}_{Initial}\right)}{D})\times 100$$
where *ML* is the larvae wet weight in grams, and *D* is the total amount of diet provided in grams.

The reduction rate (*Rr*) was calculated using the following equation:(3)$$Rr=(\frac{\left(D_{Total}-D_{Final}\right)}{D_{Total}})\times 100$$
where *D_Total_* is the total feed provided in grams (DW), and *D_Final_* is the final weight of the diet retrieved after the trials (DW).

#### Growth Rate Prediction

The growth of BSFLs, in terms of biomass production as dry weight (*X*), was predicted using the logistic equation, taking into consideration the following reasons [41]: (1) it is an equation integrated from an autocatalytic reaction mechanism based on the mass action law; (2) it is formulated with parameters of clear geometrical and biological meaning; and (3) it has been extensively and successfully used to accurately describe the growth of several microorganisms and multiple animals, including BSF larvae [42]. The algebraic form of this model is as follows:(4)$$X=\frac{X_{m}}{1+\exp\left[2+\frac{4v_{m}}{X_{m}}\left(\lambda_{X}-t\right)\right]}$$

Parameters of interest from Equation (4) were additionally calculated, with the purpose of evaluating all the characteristic growth phases of the larvae [41]:(5)$$\mu_{m}=\frac{4v_{m}}{X_{m}}$$(6)$$\tau_{x}=\lambda_{x}+\frac{2}{\mu_{m}}$$(7)$$t_{mx}=\tau_{x}+\frac{X_{m}}{2v_{m}}$$
where *X* is the growth determined; *X_m_* is the maximum growth (mg); *v_m_* is the maximum growth rate (mg d^−1^); *λ_x_* is the growth lag phase (d); *μ_m_* is the specific maximum growth rate (d^−1^); *τ_x_* is the time required to achieve half of the maximum growth (d); and *t_mx_* is the time required to reach the maximum growth (d). The experimental data of growth were processed as net dry weights, that is, the initial weights of the larvae (at time zero) were subtracted from the weight data at the different sampling times.

### 2.5. Proximal Composition

The proximate analysis of the fish substrates and BSFLs consisted of analytical determinations of crude protein, total lipid, and crude ash. Crude ash was determined by incineration at 550 °C for 4 h in a combustion oven. Total nitrogen content was determined on freeze-dried, ground samples using a CHNS elemental analyser (Leco Truspec Micro CHNS Analyzer Model 630-200-200, St. Joseph, MI, USA) and quantified according to ISO (16634-1:2008, n.d.) [43]. The crude protein content in substrates and prepupae was calculated by multiplying the total nitrogen by 6.25, according to Finke [44]. This procedure is acceptable for estimating protein content in most insect species. Energy estimation was calculated according to Codex Alimentarius [45] using the general factors for energy conversion: 17 kJ/g (4 kcal/g), 37 kJ/g (9 kcal/g), and 17 kJ/g (4 kcal/g) for protein, fat, and carbohydrate, respectively.

### 2.6. Amino Acid Content

The amino acid (AA) profile of the BSFs was quantified using a Biochrom 30 series (Biochrom Ltd., Cambridge, UK). First, the samples were hydrolysed (HCl 6 N under vacuum pressure at 110 °C for 20–24 h) and separated through a column of cation-exchange resin according to the method of Moore et al. [46]. The column eluent (lithium citrate buffer) was mixed with ninhydrin reagent and eluted at high temperature. This mixture reacted with the AAs, forming coloured compounds that were analysed at two different wavelengths: 440 and 570 nm. An internal standard of norleucine was used for quantitative analysis.

### 2.7. Lipid Content

#### 2.7.1. Reagents/Chemicals

Dichloromethane (CH_2_Cl_2_), hexane (C_6_H_14_), and methanol (MeOH) were purchased from Fisher Scientific (Leicestershire, UK). Purified water (Synergy^®^, Millipore Corporation, Billerica, MA, USA) was used whenever necessary.

#### 2.7.2. Lipid Extraction

Total lipid extraction was performed using a modified Bligh and Dyer protocol [47]. Briefly, the total lipid content was determined using representative samples of proximally 10 mg of larval powder, dry weight (DW). Ground larvae were mixed with 1 mL of Mili Q-water. Then, a solution of 1:2 (*v*/*v*) CH_2_Cl_2_:MeOH (1.25 mL:2.5 mL) was added, and the mixture was homogenised for 2 min. The glass tubes were placed on ice for 30 min and vortexed a few times for 30 s during this period. After this procedure, 1.25 mL of CH_2_Cl_2_ was added and vortexed for about 1 min, after which 1.25 mL of Mili Q-water was added and vortexed for another minute. The samples were then centrifuged (Selecta, JP Mixtasel, Abrera, Barcelona, Spain) for 5 min at 2000 rpm to separate the organic and aqueous phases. The organic phase was collected into another glass tube. This process was repeated a second time to extract the residual crude lipid in the biomass by adding 1.88 mL CH_2_Cl_2_, followed by vortexing for 1 min and centrifugation for 5 min at 2000 rpm, after which the organic phase was recovered. The lipid extracts were collected in the same tube and dried under a nitrogen stream, resuspended in 0.4 mL of CH_2_Cl_2_, vortexed, and transferred to a glass vial, previously weighed. This step was repeated twice to ensure the maximal transfer of total lipid extract to the vial. The lipid extract was dried in the vials under a nitrogen stream, and the extract content was estimated by gravimetry. The vials were stored at −20 °C before analysis by GC–MS.

#### 2.7.3. Analysis of Fatty Acid Methyl Esters (FAMEs) by Gas Chromatography–Mass Spectrometry

Fatty acid methyl esters (FAMEs) were prepared from the total lipid extracts using a methanolic solution of potassium hydroxide (2 M), according to the methodology previously described [48]. Briefly, 2 µL of a solution of hexane containing the FAMEs and 1.175 µg mL^−1^ of methyl nonadecanoate (Sigma, St. Louis, MO, USA) were used as internal standards and were analysed by gas chromatography–mass spectrometry (GC–MS). GC–MS data were acquired using an Agilent Technologies 8860 GC System (USA) equipped with a DB–FFAP column with the following specifications: 30 m long, 0.32 mm internal diameter, and 0.25 µm film thickness (123-3232, J&W Scientific, Folsom, CA, USA). The oven temperature was programmed as follows: (1) the initial temperature set up to 58 °C for 2 min; (2) a linear increase to 160 °C at 25 °C min^−1^; (3) a linear increase at 2 °C min^−1^ to 210 °C; and (4) a linear increase at 20 °C min^−1^ to 225 °C, followed by 20 min at this temperature. Helium was used as the carrier gas at a flow rate of 1.4 mL min^−1^. Five replicates were performed. The identification of each FA was performed by mass spectrum comparison with those in the NIST library. The quantitative analysis of FAs was achieved from calibration curves of each methyl ester of fatty acids from a FAME mixture (Supelco 37 Component FAME Mix, CRM47885, Sigma Aldrich, St. Louis, MO, USA), analysed by GC-MS under the same conditions of extracts, and the results expressed as mg kg^−1^ of dry biomass. The relative amounts of FAs were calculated by the ratio of the amount of each FAME and the sum of all identified FAMEs, and the results were expressed as means (%).

### 2.8. Parametric Estimations

For the growth rate analysis, fitting procedures and parametric estimations calculated from the results were carried out by minimising the sum of quadratic differences between the observed and model-predicted values, using the non-linear least squares (quasi-Newton) method provided by the macro-‘Solver’ of the Microsoft Excel spreadsheet. Confidence intervals from the parametric estimates (Student’s *t* test) and the consistency of the mathematical models (Fisher’s F test) were evaluated by the ‘SolverAid’ macro (https://learn.bowdoin.edu/excellaneous/). The significance of the comparisons between numerical parameters for each culture was analysed using one-way ANOVA followed by the Tukey test, with a significance level of *p* < 0.05.

### 2.9. Statistical Analysis

The statistical analysis were performed using one-way ANOVAs to identify statistically significant differences between BSFLs cultured under different levels of replacement of the control diet by the same by-product and compared with the control (chicken feed) (no comparisons between the use of different by-products were performed, as this was not the purpose of the present study). A Tukey post hoc test was used to assess significant differences. Significance was set at *p* < 0.05, and all the results are presented as the mean ± SD (*n* = 7, *n* = 5, or *n* = 3, depending on the dependent variables being compared).

A chemometric statistical approach (MetaboAnalyst (v5.0)) was used to find potential patterns in AAs or lipid molecular species within groups of substrates in BSFLs fed with the different tested diets. Data were log-transformed followed by auto-scaling, to decrease the influence of more and less abundant molecular species (AAs or FAs, respectively). The variance in AAs’ profiles was assessed using principal components analysis (PCA). A variable importance in projection (VIP) score was performed for the AAs. A VIP score was performed to measure the variable’s importance in the Partial Least Squares Discriminant Analysis (PLS-DA) model. The *X*-axis indicates the VIP scores corresponding to each variable on the *Y*-axis. The top variables indicate the factors with the highest VIP scores and thus the most contributory variables in class discrimination in the PLS-DA model. The coloured boxes on the right indicate the relative concentrations of the corresponding AA in each group diet studied.

A heatmap and Ward’s linkage were used to construct the dendrogram explaining the correlation between the FAs obtained in BSF larvae FA profiles and the diets supplied. Significant differences were assumed at a critical *p*-value < 0.05.

## 3. Results and Discussion

### 3.1. Performance Rates of Black Soldier Fly Larvae (BSFLs)

It is a matter of consensus that both the growth and the nutritional profile of insects are influenced by the diet supplied. No BSFLs survived when fed THH100 or CFH100; consequently, these trials are not further discussed in this study. In the case of THH 100, the amount of fat in the diet was such that it created a thick film on the top of this feeding substrate that led the larvae to drown. A recent study by Kießling et al. [49] observed that substrates with a high lipid content (>15%) led to an excessive free fat accumulation, creating anaerobic and adhesive conditions on the larval surface. These conditions impair larval respiration and hinder growth. Based on these findings, those authors concluded that a crude lipid content of approximately 10% on a dry matter basis is optimal for achieving an efficient weight gain and promoting efficient larval performance [49]. In fact, oils are used as pesticides, mostly killing insects on contact by disrupting their gas exchange (respiration) [50].

For the CFH100 trials, the viscosity was so high that the larvae were not even able to move, which ultimately caused their death as well. For the remaining feeding trials, the best survival was recorded when supplying BSFLs with CF100 and CFH25 (both reaching 99.7%), though in general, feeding trials employing CFs displayed a survival always above 98% (Table 1). For THs, the lowest recorded survival was that of THH50 (30.9%), while the best result in terms of survival was achieved using THO (95.7%). These results suggest that the fat content of the diets employed could be one of the parameters most influencing survival, as already documented by another study on the rearing of BSFLs under different diets [49]. Significant larval growth can still occur at low protein concentrations (e.g., 5%) when the crude lipid content is relatively high (e.g., approximately 12%). However, even slight variations in lipid content, such as an increase from 5% to 12%, appear to lead to noticeable reductions in weight gain [49]. These findings underscore the importance of maintaining optimal substrate properties for BSF cultivation, as inappropriate diet formulations can negatively impact larval survival, growth efficiency, and overall yield. This highlights the need for careful consideration of the diet composition in large-scale BSF production systems to ensure both productivity and animal welfare.

The bioconversion rate is commonly used as an indicator of the efficiency of BSF in bioconverting certain types of waste [51]. In this study, the maximum bioconversion efficiency was achieved with THH50 (47.2%). In fact, the bioconversion rates from THH and THO presented high scores (all above 42%), being in line with that of the control (41.6%). Considering that the diet basis consisted of chicken feed in both cases, while for THO oils extracted from THs were added to the chicken feed, our results were somewhat expected. Moreover, chicken feed is commonly used as a control, as this is known to be a diet under which BSFs can thrive with good results [52]. Lower rates of bioconversion were obtained when using CFs as a diet, namely CFH50, with the results achieved of −2.5% clearly evidencing the inability of BSFs to successfully use this diet as a nutritional source; this caveat was also reflected in the low survival of BSFs recorded under this diet. Several authors refer to the fact that the amount of protein impacts the bioconversion rate, with higher levels of protein promoting higher bioconversion values [53,54,55]. Moisture, depth of the larvae in the feeding substrate, microbial symbionts, and digestive enzymes are also known to affect bioconversion dynamics [53,56]. Therefore, it is difficult to determine a bioconversion value for a specific substrate since it is prone to fluctuations due to these different parameters. This also explains why similar substrates can present opposite bioconversion rates, as already observed by Liu et al. [56]. From an industrial point of view, these findings can have paramount importance and can aid in the selection of the best substrate to work with when scaling up production.

### 3.2. Mathematical Modelling of Black Soldier Fly Growths

Figure 2 and Figure 3 show the experimental and theoretical growth data of BSFLs under the different diets tested in this study using THs and CFs, respectively. Although the growth of the specimens being produced was measured until the final asymptotic phase, due to the end of the trial, the profiles recorded exhibited sigmoid patterns. In this way, the experimental data of larval weights were in all cases mathematically modelled using the logistic equation [3] (see above). This equation was also proposed by other authors for describing the growth of BSFLs on brewery wastes [57] and slaughtered bovine blood [58]. In another study, the logistic and the Richards equations were selected, among several non-linear models [59], as the best fitting resources to predict BSF biomass kinetics. In addition, we used the logistic model instead of Richards, since the latter is formulated with four parameters, in comparison with the three parameters which are present in the logistic model, the additional one being a shape and scaling parameter of null biological significance.

The agreement between experimental and predicted data for BSFs fed using THs was in general good and almost perfect in some cases, with a goodness of fit that varied from 0.924 to 0.999 (Table 2). The consistency of the logistic equation in predicting the growth of BSFs was corroborated for all the fittings by the F-Fisher test (*p*-values < 0.01, Table 2). The numerical parameters from this equation were statistically significant (*t*-Student test), except for most coefficients describing growth in media incorporating THs at different levels of replacement of the control diet. These statistical outcomes were better than those reported for the modelling of BSFs on mixtures of chicken feed and degassed sludge [42], in which the values of R^2^ were in the range of 0.58–0.93; due to that, the experimental data showed, in some cases, biphasic profiles not well fitted by the logistic equation.

Higher maximum larval weights (as predicted by *X_m_*) were found in media with THH and THO than when using C and THs. However, the differences between them were, on most occasions, not statistically significant (*p* > 0.05). However, the *X_m_* in THH50 was significantly higher than that recorded in the control (chicken feed) (*p* < 0.05). The value of *X_m_* (194 mg) achieved in THH50 was much higher than those obtained on food wastes (102 mg), on a mixture of food waste and bovine blood (25 mg), and using chicken feed with a 65% of water (114 mg) [58,60]. The diets formulated with 100% of THs promoted the lowest increase in larval weights, with growth even being inhibited when using THH.

These outcomes revealed the impossibility of employing a diet for BSFs based on a full replacement of the control diet by tuna by-products. Concerning the growth rate parameters *v_m_* and *μ_m_*, higher values were observed in THO10. However, when the differences recorded were compared with the C, these were not significant. Similar specific maximal growth rates, around 0.5–0.7 d^−1^, were also observed in BSFs growing on chicken feeds with a moisture content from 45% to 85% [60]. Finally, time-dependent parameters, namely *τ_x_* and *t_mx_*, recorded for THO10, also showed lower values, but these were statistically similar between the CF and C diets.

In the case of the results achieved when using CFs, the capacity of the logistic equation to describe BSF growth was somewhat less satisfactory than that achieved when using THs, with the R^2^ values recorded ranging between 0.820 and 0.923 (Table 3). This was motivated by the presence of two-phase trends, as described by diauxic growth in the microbiology [61], and which could be modelled mathematically with the sum of two logistic equations [41]. No parametric data for CFH100 and CFH50 were compiled, as the feeds formulated with these percentages of control diet replacement by CFH were not valid and even prevented larval growth. The best *X_m_* value (134.1 mg) was recorded for C but had no significant differences to that reported by CF25 and CF50 (*p* > 0.05). This lack of significant differences between diets was also found for the remaining time- and rate-dependent parameters calculated using the modelling approach employed in the present study.

To select the best substrate candidate, we calculated the simple linear correlation between growth responses (survival, bioconversion efficiency, and maximum growth—*X_m_*) and levels of replacement (Appendix A). For THs, the level of 50% is the compromise option, resulting in the highest survival, bioconversion, and interesting biomass production (slightly less in THH100). For THH and CFH, only two levels could not be linearised, and the lowest percentages directly showed the best performances (25% in both cases). In this context, the lowest incorporations of tuna oil and cod frame by-products (THO2.5 and CF25) also proved to be the most adequate substrates for BSF growth. Based on these results, the best options in terms of the maximum values of the mentioned growth variables were THO2.5 and THH25.

### 3.3. Proximate Composition of Fish By-Products

The proximate composition of the raw materials used, both untreated and hydrolysed, from the two by-products is summarised in Table 4. The moisture content in all the dry materials was lower than 2%, and the presence of inorganic ingredients (such as ash) was, in all cases, at least 15%. The type of codfish by-product used (frames) was the source of the highest level of ash content observed among the feeding substrates tested (30.4%). The high ash values in the hydrolysates free of bones (15–19%) are most likely due to the alkalis added during the digestion process to maintain the pH at levels that maximise the hydrolytic capacity of the alcalase. More than 80% of the weight was organic matter, except for the CFs. These values are consistent with those reported for the production of tailored hydrolysates from blue whiting discards [62], in which organic matter and ash content were around 78–81% and 13–15%, respectively.

The diets used in the trials from both by-products (hydrolysed, untreated, or partially blended with chicken feed as the control) presented several variations (Table 5). Concerning untreated tuna heads (THs), including pure (TH100) and blended with chicken feed (TH50 and TH25), it is possible to observe that as the content in chicken feed increased, the amounts of ash, crude protein, and crude lipid decreased. This is consistent with the composition of chicken feed (C), which has remarkably lower levels of ash, protein, and lipid compared with TH. Diets with a higher proportions of chicken feed also resulted in increased energy levels.

The same pattern was observed with both untreated or hydrolysed codfish frames, whether pure or blended with increasing percentages of chicken feed (C). Regarding ash levels, the highest value recorded was for CF100 with 28.9%. This can be explained by the high content of fish bones (frames) and the alkalis added during the digestion process. Chicken feed blended with 2.5% of tuna head oil (THO2.5) showed the lowest ash content (5%), which is similar to the percentage recorded for C (chicken feed) (5.5%). For crude protein, TH100 and CF100 displayed the highest values, at 50.8% and 50.6%, respectively. In contrast, the feeds formulated with chicken feed (THO10, THO5, THO2.5, and C) exhibited the lowest levels of crude protein (15.4%, 20.7%, 17.9%, and 17.2%, respectively). Additionally, it is possible to observe that blending TH, THH, CF, and CFH with chicken feed (C) decreases the protein levels. A similar pattern was observed for crude lipid, with TH100, THH25, and THH50 showing maximum values of 28.9%, 24.8%, and 20.5%, respectively, and C presenting the lowest level recorded with 2.9%.

### 3.4. Proximate Composition of Black Soldier Fly Larvae

The ash content (%) of the BSFLs ranged from 21.7% (C) to 4.6% (CFH50), which is within the regular values reported in the literature for this species [63]. In general, a pattern can be associated with the feed trials and their different percentages of replacement of the control diet. The main differences are related to the stage of maturity of the larvae. The trials where the larvae entered in pre-pupae earlier presented a higher ash content. This is due to the increase in the exoskeleton and its calcium content.

In the case of crude protein (%), CF100 showed the highest level with 58.6%, and CFH25 displayed the lowest with 32.1%. These results are in line with previous tests [64] and demonstrate the adequacy of BSFL biomass as an ingredient for aquafeed production.

### 3.5. Amino Acid Profile of Black Soldier Fly Larvae

In general, protein and AA requirements for fish farming are well established and tailored to each fish species, aiming to maximise growth and health [65]. The protein content in aquafeeds generally averages 30–35% for marine shrimp, 28–32% for catfish, 35–40% for tilapia, 38–42% for hybrid striped bass, and 40–45% for trout and other marine finfish [66]. When comparing the crude protein content of the diets tested with that of the BSFs, it is possible to observe that higher protein content in the feed generates larvae with higher crude protein levels. Moreover, our results show that the protein content of BSFs, when supplied with all the tested diets, fulfils the protein requirements for fish aquafeeds, ranging from 32.1% with CFH25 to 58.6% with CF100. Studies evaluating diets containing BSF meal in Atlantic salmon (*Salmo salar*) across a complete production cycle in sea cages have shown no significant differences in general fillet parameters when compared to salmon fed commercial diets. These findings confirm the feasibility of incorporating BSF meal into salmon diets at inclusion levels of up to 10%, supporting its use as a sustainable protein source in aquaculture without compromising product quality or performance [67].

The requirements for AAs are similar for all fish species: arginine, histidine, isoleucine, leucine, lysine, methionine, phenylalanine, threonine, tryptophan, and valine are essential for normal growth and metabolic function [68,69]. Thus, to maintain a healthy fish production, it is crucial to include these 10 AAs, which should account for at least 40–60% of the total protein in the diet. Regarding the results obtained with BSFs, 18 AAs were recorded, with all the essential AAs listed above present in their biochemical profile. The proportion of essential AAs is generally higher than 40%, especially when using THs, with a maximum of 45.5% being recorded for TH50. A similar trend was recorded for BSFLs reared on CFs, where this proportion ranged from a minimum value of 38.7% with CFH50 to a maximum of 43.9% with CF25.

Overall, when considering the AA content, few differences can be observed when comparing the larvae reared on the control diet and those reared on substrates containing tuna heads (THs) (Table 6). Additionally, hydrolysis did not result in substantial differences in the content of most AAs for THs. The AAs recorded in the highest percentages for the BSFs were glutamic and aspartic acids. In contrast, cysteine, hydroxyproline, and methionine displayed the lowest percentages across all the diets tested. In the case of BSFs reared on CFs (Table 6), the relative abundance of threonine, alanine, methionine, isoleucine, tyrosine, phenylalanine, and lysine was higher than those found for BSFs reared on the control diet. Interestingly, the percentage of glutamic acid was drastically halved when BSFs were reared with CFs compared to the control.

In modern aquaculture, soymeal is used to replace fishmeal in aquafeed formulations [70]. Despite the AA profile of soymeal being generally of better quality than that of other plant-based ingredients, it is still deficient in lysine, methionine, threonine, and valine when compared to animal-based protein sources [71,72]. Previous studies established that the AA profile of BSF is comparable to that of soybean meal, with methionine and lysine being the most limiting essential AAs for fish production [71]. The AA profile of BSFs reared on several organic substrates has a higher quality than the standard AA profiles reported for soybean and sunflower meal [73]. The methionine and lysine levels observed in BSF larvae in this study surpassed the ranges recommended for fishmeal as outlined by the National Research Council (NRC) standards for aquaculture species [74]. Specifically, the methionine content (1.5–2.4%) in BSFLs exceeded the 0.5–0.9% range reported for BSFs reared on alternative by-products in previous studies [75]. Similarly, the lysine levels recorded (5.8–8.4%) were significantly higher than the 2.1% reported by Gutiérrez et al. [76] for BSFs reared on chicken manure. Additionally, research on BSF meal indicates that its inclusion as a replacement for fish meal can positively influence the hypocholesterolaemic/hypercholesterolaemic ratio and enhance the levels of essential amino acids and microelements [77]. In studies involving gilthead sea bream (*Sparus aurata*) juveniles, the dietary inclusion of up to 45% BSF meal, effectively replacing 100% of fish meal protein, demonstrated no adverse effects on immune function or oxidative status. Furthermore, BSF meal-based diets were found to improve the intestinal antioxidant status of the juveniles [78]. These findings highlight the remarkable nutritional potential of BSF as a protein source, further supported by its demonstrated suitability as an alternative feed ingredient for aquaculture diets, with, also, no observed negative impacts on growth performance or overall health, for the like of white Pacific shrimp (*Litopeneaus vannamei*), sea bream (*Sparus arauta*), or European sea bass (*Dicentrarchus labrax*) [79,80,81].

The principal component analysis (PCA) score plot displayed in Figure 4, where the two principal components explain more than 50% of the variability recorded, reveals few differences in the AA profile of BSFs reared on the different diets provided. However, THH50 stands out, indicating that the AA profile of BSFs reared with this diet was somewhat different. Ellipses around the groups indicate variance within each treatment, with diets such as CF100 and TH100 showing tighter clustering, signifying more uniform amino acid profiles under these treatments. In contrast, treatments with mixed or lower inclusion levels exhibited greater variability, as shown by broader ellipses. A VIP (variable importance in projection) score of the AAs retrieved from the feeding trials is presented in Figure 5. The relative concentrations of larval AAs presented in Figure 5 demonstrate that there is no clear pattern or correlation between types of by-products, treatments, or percentages of inclusion and the AA concentration in the larvae. However, valine, isoleucine, and leucine displayed higher concentrations with THs. For methionine and lysine, critical limiting AAs in aquafeeds, the CF diets (CF100, CF25, CF50, and CFH50) proved to be the most significant contributors.

Overall, the different diets tested resulted in the production of a high-quality protein source comparable to fish meal and clearly superior to many the plant-based protein sources commonly used in aquafeed formulations.

### 3.6. Fatty Acid Profile of Black Soldier Fly

For lipids, the total lipid content typically accounts for 7–15% of aquafeed formulations. The results obtained in our study for the total lipid content showed a similar pattern recorded for crude protein, with the tested diets presenting higher lipid levels, leading to BSF larvae also with correspondingly higher lipid levels. Moreover, the amount of crude fat content recorded met the requirements commonly associated with the production of aquafeed [82]. The values recorded, summarised in Table 4, ranged from 15.3% (CF100) to 35.3% (CFH25). Based on the FA analysis (Table 7), 15 different FAs were recorded in the insect’s lipid fraction (Table 4). Across all treatments, the following FAs were consistently present: FA 12:0, FA 14:0, FA 16:0, FA 16:1, FA 18:0, FA 18:1, FA 18:2 *n*-6, FA 18:3 *n*-3, and FA 22:6 *n*-3. Saturated fatty acids (SFAs) were the predominant class, followed by monounsaturated fatty acids (MUFAs) and finally polyunsaturated fatty acids (PUFAs). While SFAs are the predominant class in the lipid profile of BSF, studies involving gilthead sea bream (*Sparus aurata*) have demonstrated that diets supplemented with BSF-derived oils effectively replace vegetable oils without adversely affecting fillet composition or quality attributes [83]. Among the SFAs, the most abundant was FA 16:0 (palmitoleic acid) (19.2–27.6%), followed by FA 18:0 (stearic acid) (6.1–20.8%). Lauric acid (FA12:0), which usually represents a major FA on BSF lipid profiles, in this study displayed relative abundances ranging from 0.61 to 25.58%, lower levels than those reported in previous studies on BSF [64]. This finding is likely explained by the nutritional profile of the diets tested in the present trial, where lauric acid is completely absent (Appendix A). Lauric acid (C12:0) exhibits antimicrobial properties and modulates the immune response in marine fish, enhancing their resistance to pathogens by promoting the production of pro-inflammatory cytokines and supporting gut health [84,85]. Studies have demonstrated that diets incorporating BSF can improve fish development, antioxidative capability, gut microbiota, and intestinal health [86,87]. Additionally, lauric acid is well documented for its antibacterial and antiviral activity, contributing to the immunomodulating capacity of fish [88,89]. Thus, incorporating BSF into aquafeeds can improve immune function and promote the overall health of marine fish species. In the present study, BSFs incorporated higher levels of PUFAs when supplied with THH50, THO10, THO5, THO2.5, and CFH50 (26.2%, 24.9%, 26.6%, 22.9%, and 23.8%, respectively). These results align with those obtained by Liland et al. [31] for brown algae powder (18–25%) but exceed the findings obtained by Barroso et al. [90] (10–18%) and St-Hilaire et al. [32] (3–4%) for fish wastes. Hoc et al. [26] tested diets enriched with flaxseed marked with deuterated water. Flaxseed is recognised for its high Linolenic acid content. Their study revealed an absence of deuterated PUFAs in the FA profile of BSF, leading to the conclusion that BSF lacks the capacity to biosynthesise these types of FAs. This is also supported by Duarte et al. [91], who investigated BSF lipid metabolism using deuterium-labelled water and nuclear magnetic resonance (^2^H-NMR) to trace lipid synthesis pathways and concluded that, despite BSFLs being able to incorporate fatty acids from macroalgae, they had a limited ability for the de novo synthesis of specific PUFAs.

These findings suggest that the higher bioavailability of these biomolecules likely increases the ability of BSFLs to incorporate them in their body composition. Conversely, BSFs fed the diets C and TH100 showed the lowest levels of PUFAs, at 9.9% and 7.7%, respectively. In the case of TH100, the high diet viscosity restricted BSF mobility, reducing their metabolism and extending their life cycle. Liu et al. [56] reported that limitations in media can extend the life cycle of BSFs to 70 days. Consequently, the larvae used for FA analysis were not on the same development stage as conspecifics fed with the other diets and may not have incorporated as much FA as other BSFLs reared using diets with a lower percentage of THs in their composition. Regarding C, the lower content of PUFAs in this diet naturally led to a lower percentage of PUFAs in the larvae profile. The highest levels of α-Linolenic acid (ALA, FA 18:3 *n*-3) (7.3%), FA 20:5 *n*-3 EPA (6.7%), and FA 22:6 *n*-3 DHA (7.1%) were obtained when using diets CFH50 (for ALA) and THO5 (for both EPA and DHA), featuring significantly higher levels than those recorded in C (where ALA accounted for 0.42% of the total pool of FA, EPA was absent, and DHA only represented 0.30%).

When considering the inclusion of BSF as an insect-based ingredient for marine fish, one needs to take into consideration the PUFA nutritional requirements displayed by these organisms, namely in long-chain PUFAs from the *n*-3 family [92]. These requirements can be successfully fulfilled by using a BSF insect meal such as the one produced with the present work, in line with the findings reported by Newton [12] and Glencross [5]. Moreover, the total levels of *n*-3 PUFA peaked at 14.41% for BSFs fed with THO5, a noteworthy increase when compared to C, which only displayed 0.72%. Our results confirm the successful incorporation of considerable amounts of *n*-3 PUFA in BSF. Similar results were obtained by Arena et al. [93], where diets incorporating fish by-products led to a substantial increase in the levels of EPA and DHA in BSF larvae and pre-pupae. This highlights that utilising fish processing by-products is an effective strategy to enhance the nutritional quality of BSF as a component in aquafeeds, offering a sustainable method to improve the dietary value of insect-based ingredients. However, our study shows that high levels of PUFA in diets (Appendix A) do not necessarily lead to high levels of the incorporation of these biomolecules by BSF, as several other parameters, such as moisture, viscosity, or molecular bioavailability, are also paramount to promoting a successful incorporation of PUFA into their biomass. Encouraging results have been demonstrated with feeding trials using BSFs enriched in ALA + EPA + DHA in the diet of Nile tilapia, where LC-PUFA-enriched BSF meal in association with chitinase was considered as an effective alternative to fishmeal, improving protein digestion processes, stimulating beneficial microbiota, and ultimately fish growth rates [94].

The heatmaps exhibited in Figure 6 show the correlation between the different FAs and the lipid profile of BSFs compared with the different diets used in the present study. Additionally, it also displays a dendrogram where a clear separation of CFs from THs can be observed, indicating that these two different substrates generated different lipid profiles in BSFs. The control diet was considered to promote the production of BSFLs with a more similar FA profile to that of CFs. This separation occurs mainly due to the presence of FA 17:0, FA 15:0, and FA 17:1, which possess a high correlation with THs. Also, FA 20:4 *n*-6, FA 22:6 *n*-3, and FA 14:0 show a high correlation with THs when compared to CFs.

Thus, the complete biochemical profile of the larvae produced in this study is within the nutritional demands of other farmed animal species, such as pigs and poultry [95,96,97]. Also, BSFLs can represent a valuable source of nutrients for humans and were already approved by the EU (Reg. (UE) 2015/2283), with the biomass of the BSFLs produced in this study reinforced with a valuable content of essential FAs (e.g., ALA, EPA, and DHA), which are known to positively impact human health [98].

## 4. Conclusions

This study establishes a baseline for evaluating the efficiency of BSF in incorporating essential fatty acids (particularly PUFAs) from fish processing by-products and co-products (tuna canning and codfish salting) using untreated or hydrolysed substrates, as well as extracted oils. The goal was to produce an insect-based biomass for aquafeed formulations, especially for marine species. The results show that most of the diets tested supported BSF production with adequate levels of lysine and methionine, particularly with THH performing best for lysine and CFs for methionine. Notably, THO5, which incorporated 5% oil, yielded over 25% PUFAs, including the highest levels of EPA and DHA. However, BSFs did not survive in extreme diet replacements (at the highest levels of THH and CFH incorporated), emphasising the importance of preliminary trials to determine appropriate replacement levels. Factors such as texture, moisture, and nutrient bioavailability influenced the metabolic performance and biochemical profile of the BSFs. This study highlights the potential of BSF to upcycle by-products, contributing to circular economy goals. Furthermore, these findings align with life cycle assessments, demonstrating that insects become environmentally beneficial when reared on waste streams. This work advances the validation of THs and CFs as sustainable BSF diets, promoting more sustainable ingredient options for aquafeeds.

## Figures and Tables

**Figure 1 insects-16-00113-f001:**
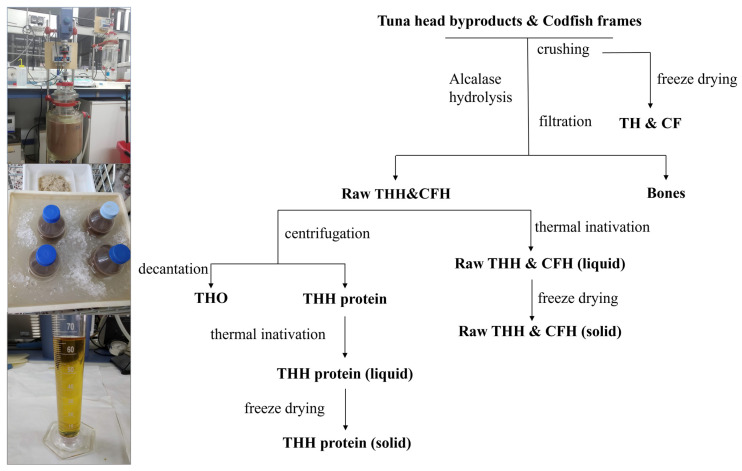
Flowchart describing the production of hydrolysates and oil from tuna heads and from codfish frames employed to culture black soldier fly larvae. (TH—untreated tuna head; THH—tuna head hydrolysate; THO—tuna head oil; CF—untreated codfish frame; CFH—codfish frame hydrolysate).

**Figure 2 insects-16-00113-f002:**
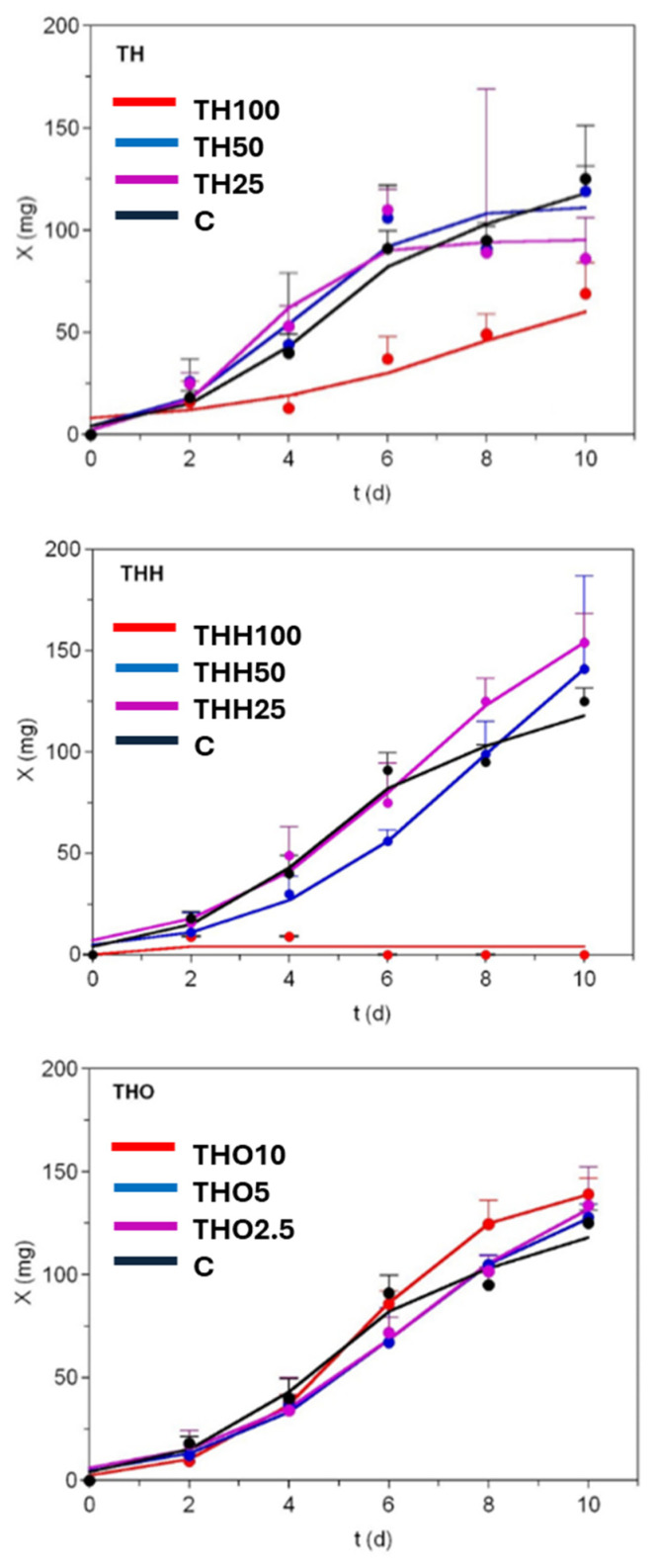
Growth of black soldier fly supplied with tuna heads at different replacement levels. TH stands for untreated tuna head, THH for tuna head hydrolysate, and THO for tuna head oil. Experimental data were always fitted to the logistic equation (continuous lines). Only the superior half of the error bars is shown (standard deviation) for clarity. This was calculated using the data from 10 individuals per 5 replicates for each substrate.

**Figure 3 insects-16-00113-f003:**
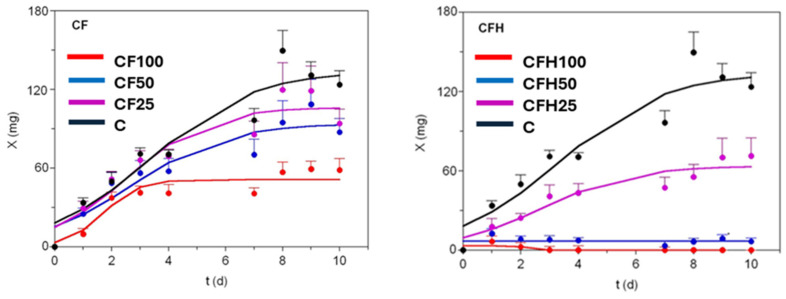
Growth of black soldier fly supplied with codfish frames at different replacement levels. CF stands for untreated codfish frame and CFH stands for codfish frame hydrolysate. Experimental data were always fitted to the logistic equation (continuous lines). Only the superior half of the error bars is shown (standard deviation) for clarity. This was calculated using the data from 10 individuals per 7 replicates for each substrate.

**Figure 4 insects-16-00113-f004:**
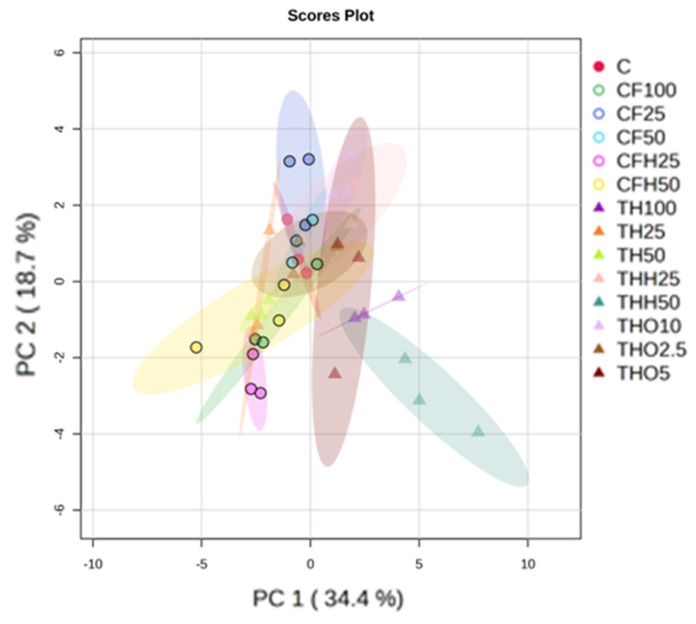
Principal component analysis score plot of the amino acid profile of black soldier fly larvae supplied with different feeding substrates derived from tuna heads (TH: untreated tuna head, THH: tuna head hydrolysate, THO: tuna head oil) and codfish frames (CF: untreated codfish frame, CFH: codfish frame hydrolysate) at different replacement levels of the control diet (C: chicken feed).

**Figure 5 insects-16-00113-f005:**
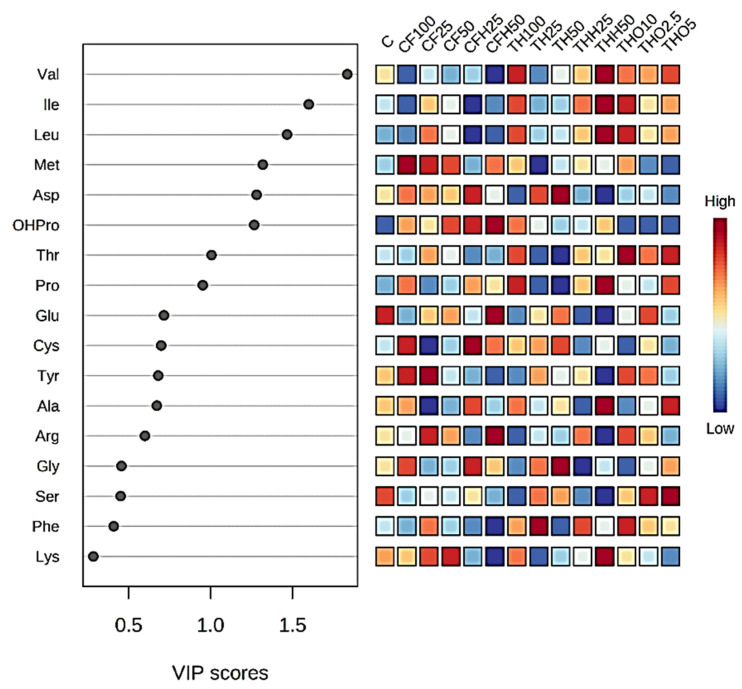
VIP scores of the amino acid content of black soldier fly larvae supplied with different feeding substrates derived from tuna heads (TH: untreated tuna head, THH: tuna head hydrolysate, THO: tuna head oil) and codfish frames (CF: untreated codfish frame, CFH: codfish frame hydrolysate) at different replacement levels of the control diet (C: chicken feed).

**Figure 6 insects-16-00113-f006:**
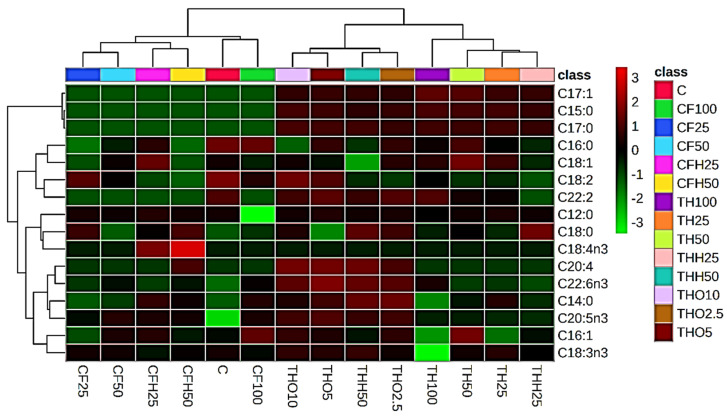
Heatmap of the fatty acid content of black soldier flies supplied with different feeding substrates derived from tuna heads (TH: untreated tuna head, THH: tuna head hydrolysate, THO: tuna head soil) and codfish frames (CF: untreated codfish frame, CFH: codfish frame hydrolysate) at different replacement levels of the control diet (C: chicken feed).

**Table 1 insects-16-00113-t001:** Survival, initial and final weight, bioconversion efficiency (%), and reduction rate (%) of black soldier fly larvae supplied with different replacement levels of tuna heads (untreated—TH, hydrolysates—THH, and oil—THO) and codfish frames (untreated—CF and hydrolysates—CFH). Errors correspond to the confidence intervals for *n* = 7 (replicates of processing) and α = 0.05.

Trials	Survival (%)	Initial Larval Weight (mg)	Final Larval Weight (mg)	Bioconversion Efficiency (%)	Reduction Rate (%)
**TH100**	63.4 ± 11.0	12.0 ± 0.01	80.9 ± 0.01	22.98 ± 5.5 ^a^	29.9 ± 15.8
**TH50**	90.6 ± 7.3 ^a^	15.4 ± 0.02	134.8 ± 0.03	39.79 ± 12.3 ^b^	65.1 ± 1.9
**TH25**	91.1 ± 6.2 ^a^	11.6 ± 0.01	97.2 ± 0.02	28.54 ± 6.6 ^a^	83.5 ± 0.8 ^b^
**THH50**	30.9 ± 5.5	6.4 ± 0.00	147.9 ± 0.05	47.16 ± 5.4 ^b^	34.2 ± 9.9
**THH25**	75.1 ± 9.7	5.1 ± 0.01	159.0 ± 0.01	51.31 ± 4.7 ^b^	75.2 ± 2.4
**THO10**	95.7 ± 2.9 ^ab^	5.2 ± 0.00	144.3 ± 0.01	46.35 ± 3.0 ^b^	81.7 ± 3.5 ^ab^
**THO5**	93.1 ± 2.5 ^a^	5.1 ± 0.00 ^b^	132.9 ± 0.01	42.60 ± 2.3 ^b^	82.7 ± 1.7 ^ab^
**THO2.5**	94.6 ± 3.1 ^a^	4.6 ± 0.00	138.1 ± 0.02	44.61 ± 6.32 ^b^	83.0 ± 0.9 ^ab^
**CF100**	99.7 ± 0.7 ^a^	25.9 ± 0.07 ^a^	85.2 ± 0.09	14.6 ± 1.7	24.8 ± 2.1
**CF50**	99.1 ± 1.6 ^ab^	25.7 ± 0.09 ^a^	112.3 ± 0.10	28.4 ± 3.6 ^a^	54.3 ± 3.7
**CF25**	98.6 ± 2.5 ^ab^	24.4 ± 0.80 ^a^	118.1 ± 0.10	35.5 ± 2.5 ^a^	67.7 ± 2.7
**CFH50**	34.0 ± 17.4	18.0 ± 0.07	23.5 ± 0.03	−2.5 ± 1.0	5.5 ± 4.1
**CFH25**	99.7 ± 0.8	22.5 ± 0.12	94.7 ± 0.13	21.7 ± 4.7 ^a^	48.8 ± 5.4
**C**	94.6 ± 3.4 ^b^	5.1 ± 0.00 ^b^	130.1 ± 0.01	41.65 ± 2.4 ^b^	85.4 ± 1.0 ^b^

Note: Values with different letters display significant differences by Tukey’s post-hoc test for a significance of *p* < 0.05. Comparisons within different percentages of the same diet.

**Table 2 insects-16-00113-t002:** Numerical values and confidence intervals for parameters derived from logistic equation [3] applied for growth of black soldier fly larvae supplied with different levels of replacement of tuna heads (TH—untreated tuna head, THH—tuna head hydrolysate, and THO tuna head oil). R^2^ is determination coefficient between experimental and predicted data. Consistency of fittings was also determined (*p*-value from F-Fisher test). NS: not significant. Different letters in each column indicate significant difference between wastes (*p* < 0.05).

Parameters	TH100	TH50	TH25	THH50	THH25	THO10	THO5	THO2.5	C
***X_m_* (mg)**	135.5 (NS)	112.4 ± 44.1 ^a^	95.1 ± 33.3 ^a^	194.0 ± 34.3 ^b^	181.8 ± 68.9 ^ab^	143.5 ± 7.0 ^a^	142.3 ± 24.2 ^ab^	154.5 ± 46.7 ^ab^	121.5 ± 39.0 ^a^
***v_m_* (mg d^−1^)**	75.7 (NS)	22.5 (NS)	25.6 (NS)	22.8 ± 3.2 ^a^	22.3 ± 7.1 ^a^	26.6 ± 2.9 ^a^	19.8 ± 3.9 ^a^	19.2 ± 4.9 ^a^	20.3 ± 15.5 ^a^
***λx* (d)**	13.98 (NS)	1.61 (NS)	1.56 (NS)	3.64 ± 0.74 ^a^	2.42 ± 1.32 ^a^	2.76 ± 0.32 ^a^	2.54 ± 0.73 ^a^	2.44 ± 1.06 ^a^	1.89 (NS)
***μ_m_* (d^−1^)**	0.388 (NS)	0.799 (NS)	1.08 (NS)	0.469 ± 0.064 ^a^	0.494 ± 0.282 ^ab^	0.743 ± 0.103 ^b^	0.556 ± 0.173 ^b^	0.498 ± 0.227 ^ab^	0.667 ± 0.641 ^ab^
***τ_x_* (d)**	7.72 (NS)	4.11 ± 2.21 ^a^	3.41 ± 1.93 ^a^	7.90 ± 2.19 ^a^	6.47 ± 2.11 ^a^	5.45 ± 0.24 ^a^	6.14 ± 0.90 ^a^	6.46 ± 1.66 ^a^	4.88 ± 1.87 ^a^
***t_mx_* (d)**	29.6 (NS)	6.61 ± 5.02 ^a^	5.27 ± 3.99 ^a^	12.2 ± 2.9 ^a^	10.5 ± 4.2 ^a^	8.14 ± 0.54 ^a^	9.74 ± 1.90 ^a^	10.5 ± 3.3 ^a^	7.88 ± 4.21 ^a^
**R^2^**	0.949	0.939	0.924	0.998	0.992	0.999	0.998	0.995	0.974
***p*-values**	<0.01	<0.01	<0.01	<0.001	<0.001	<0.001	<0.001	<0.001	<0.001

**Table 3 insects-16-00113-t003:** Numerical values and confidence intervals for parameters derived from the logistic equation [3] applied for the growth of black soldier fly larvae supplied with different levels of replacement of codfish frames (CF—untreated codfish frame and CFH—codfish frames hydrolysate). R^2^ is the determination coefficient between experimental and predicted data. Consistency of the fittings was also determined (*p*-value from F-Fisher test). NS: not significant. Different letters in each column indicate a significant difference between wastes (*p* < 0.05).

Parameters	CF100	CF50	CF25	CFH25	C
***X_m_*** **(mg)**	51.3 ± 9.4 ^a^	94.0 ± 23.5 ^b^	106.4 ± 21.5 ^b^	63.7 ± 14.4 ^c^	134.1 ± 38.9 ^b^
***v_m_* (mg d^−1^)**	20.5 (NS)	14.2 ± 13.1 ^a^	18.7 ± 16.1 ^a^	10.2 ± 9.0 ^a^	18.5 ± 15.2 ^a^
***λx* (d)**	0.456 (NS)	−0.598 (NS)	−0.274 (NS)	−0.363 (NS)	−0.276 (NS)
***μ_m_* (d^−1^)**	1.60 (NS)	0.603 (NS)	0.704 ± 0.666 ^a^	0.641 ± 0.639 ^a^	0.551 (NS)
***τ_x_* (d)**	1.71 ± 0.84 ^a^	2.72 ± 1.71 ^a^	2.57 ± 1.31 ^a^	2.76 ± 1.53 ^a^	3.35 ± 2.06 ^a^
***t_mx_* (d)**	2.96 ± 1.77 ^a^	6.04 ± 4.67 ^a^	5.41 ± 3.43 ^a^	5.88 ± 4.09 ^a^	6.98 ± 5.34 ^a^
**R^2^**	0.923	0.820	0.891	0.857	0.861
** *p* ** **-values**	<0.01	<0.001	<0.001	<0.001	<0.001

**Table 4 insects-16-00113-t004:** Proximate composition (as %, *w*/*w* of initial material) of tuna heads and codfish frames (THs and CFs) and their respective hydrolysates (THH and CFH) in terms of moisture (Mo), organic matter (OM), and ash content (Ash). Total lipids (Lip), proteins (Pr-tN, as total nitrogen ×6.25), and proteins after degreasing samples (Pr-tN*). TEAA/TAA is the ratio as a percentage between essential and total amino acids. Errors indicate the confidence intervals for *n* = 4 and α = 0.05.

Substrates	Mo (%)	Ash (%)	OM (%)	Lip (%)	Pr-tN (%)	TEAA/TAA (%)
**TH**	0.3 ± 0.2	15.7 ± 0.2	84.1 ± 0.2	31.7 ± 0.6	54.2 ± 1.4	38.71 ± 0.80
**CF**	0.5 ± 0.4	30.4 ± 0.4	69.2 ± 0.6	5.3 ± 0.7	58.9 ± 6.8	27.48 ± 0.76
**THH**	1.7 ± 0.4	15.5 ± 3.9	82.9 ± 3.7	32.1 ± 2.2	45.4 ± 1.0	45.55 ± 0.78
**CFH**	0.8 ± 0.3	18.7 ± 0.1	80.5 ± 0.3	2.1 ± 0.3	74.8 ± 3.3	40.61 ± 0.79

**Table 5 insects-16-00113-t005:** Ash, crude protein, and total lipid content and energy (kJ/g) of diets and black soldier flies supplied with different levels of replacement of tuna heads (TH: untreated tuna head, THH: tuna head hydrolysates, THO: tuna head oil) and different levels of codfish frames (CF: untreated codfish frame, CFH: codfish frame hydrolysate) (g/100 g total of dry weight) (mean ± SD; *n* = 7 for lipid, *n* = 3 for protein, and *n* = 5 for ash).

	Samples	Ash	Protein	Lipid	Energy (kJ/g)
Diets—Levels of replacement	**TH100**	16.23 ± 0.56 ^a^	50.84 ± 0.22 ^a^	28.94 ± 1.61 ^a^	12.5
	**TH50**	8.07 ± 1.11	36.29 ± 0.66 ^b^	16.49 ± 2.85 ^bc^	20.6
	**TH25**	6.69 ± 1.76	30.55 ± 0.75 ^c^	9.60 ± 3.00 ^c^	24.2
	**THH50**	7.85 ± 0.82	37.04 ± 1.42 ^b^	20.52 ± 2.36 ^bd^	19.4
	**THH25**	7.07 ± 0.48	42.76 ± 4.98 ^d^	24.82 ± 2.07 ^d^	16.4
	**THO10**	6.19 ± 1.12	15.39 ± 1.51 ^e^	9.57 ± 1.52 ^c^	29.9
	**THO5**	6.38 ± 2.66	20.75 ± 1.08 ^f^	7.23 ± 0.74 ^ce^	28.3
	**THO2.5**	5.00 ± 4.31	17.92 ± 0.16 ^g^	4.75 ± 1.56 ^ef^	29.9
	**CF100**	28.96 ± 3.36 ^b^	50.64 ± 1.32 ^a^	5.20 ± 1.01 ^e^	17.5
	**CF50**	18.85 ± 2.55 ^a^	38.25 ± 2.02 ^b^	3.42 ± 0.25 ^f^	22.6
	**CF25**	7.02 ± 1.00	30.23 ± 0.79 ^c^	6.88 ± 0.41 ^ec^	24.9
	**CFH50**	17.88 ± 0.89 ^a^	41.28 ± 1.13 ^d^	5.98 ± 0.98 ^ce^	20.9
	**CFH25**	6.66 ± 2.23	28.66 ± 0.88 ^c^	7.22 ± 1.34 ^c^	25.4
	**C**	5.54 ± 2.23	17.17 ± 0.35 ^g^	2.87 ± 1.08 ^f^	30.6
Black soldier fly larvae	**TH100**	13.06 ± 0.75 ^ab^	54.11 ± 1.01 ^a^	23.03 ± 4.04	
	**TH50**	17.49 ± 3.66 ^ac^	47.16 ± 1.78	27.50 ± 1.92	
	**TH25**	12.48 ± 0.97 ^b^	46.01 ± 0.62	24.34 ± 2.72	
	**THH50**	17.54 ± 3.52 ^ac^	44.89 ± 0.87	24.20 ± 4.00	
	**THH25**	17.39 ± 3.60 ^ac^	45.30 ± 2.06	25.46 ± 2.13	
	**THO10**	13.29 ± 2.15 ^ab^	40.77 ± 0.16 ^b^	30.18 ± 2.27 ^a^	
	**THO5**	19.53 ± 3.66 ^c^	43.52 ± 2.87	22.62 ± 2.65	
	**THO2.5**	18.95 ± 2.70 ^c^	44.60 ± 1.12	21.93 ± 3.06	
	**CF100**	14.18 ± 4.36 ^ac^	58.60 ± 1.09 ^c^	15.31 ± 5.38 ^b^	
	**CF50**	16.38 ± 2.38 ^ac^	33.42 ± 6.88 ^d^	20.76 ± 6.05	
	**CF25**	20.15 ± 3.46 ^c^	32.90 ± 7.39 ^d^	23.84 ± 3.54	
	**CFH50**	4.61 ± 3.12 d	55.96 ± 7.39 ^a^	19.95 ± 7.28	
	**CFH25**	13.18 ± 9.34	32.11 ± 1.60 ^d^	35.35 ± 5.49 ^a^	
	**C**	21.74 ± 4.88 ^c^	45.04 ± 1.44	21.53 ± 2.46	

Note: Values with different letters display significant differences by Tukey’s post hoc test for a significance of *p* < 0.05. Comparisons within different percentages of the same diet.

**Table 6 insects-16-00113-t006:** Amino acid content of black soldier fly larvae (% or g/100 g total amino acids) supplied with different replacement levels of tuna heads (TH: untreated tuna head, THH: tuna head hydrolysate, THO: tuna head oil) and codfish frames (CF: untreated codfish frame, CFH: codfish frame hydrolysate). Errors are the confidence intervals for *n* = 3 (replicates of processing) and α = 0.05. TEAA/TAA is the ratio as a percentage between essential and total amino acids.

Amino Acids	TH25	TH50	TH100	THH25	THH50	THO2.5	THO5	THO10	CF25	CF50	CF100	CFH25	CFH50	C
**Asp**	11.15 ± 0.93 ^a^	12.32 ± 0.32 ^a^	8.33 ± 0.50	9.54 ± 0.26 ^b^	8.13 ± 0.57	9.87 ± 0.84 ^ab^	9.28 ± 0.56 ^ab^	9.75 ± 0.78 ^ab^	10.76 ± 0.90 ^ab^	10.64 ± 0.34 ^ab^	11.01 ± 0.63 ^ab^	12.03 ± 0.63	9.86 ± 0.40	10.31 ± 0.36 ^b^ 4.19 ± 0.05 ^b^
**Thr**	3.46 ± 0.61 ^a^	3.42 ± 0.13 ^a^	4.49 ± 0.35 ^b^	4.32 ± 0.48 ^ab^	4.31 ± 0.20 ^ab^	4.36 ± 0.40 ^ab^	4.52 ± 0.22 ^ab^	4.76 ± 0.36 ^a^	4.31 ± 0.35 ^ab^	4.21 ± 0.18 ^ab^	3.76 ± 0.26 ^a^	3.45 ± 0.08 ^a^	3.58 ± 0.24 ^a^	
**Ser**	5.40 ± 0.11 ^a^	5.30 ± 0.21 ^a^	4.70 ± 0.27	4.78 ± 0.60 ^a^	4.65 ± 0.71 ^a^	5.64 ± 0.19 ^ab^	5.80 ± 0.58 ^ab^	5.28 ± 0.20 ^a^	5.09 ± 0.35 ^a^	5.05 ± 0.32 ^a^	4.98 ± 0.23 ^a^	5.27 ± 0.14	4.83 ± 0.23	5.64 ± 0.07
**Glu**	12.32 ± 1.25 ^ab^	12.46 ± 0.53 ^ab^	11.11 ± 1.64 ^a^	11.04 ± 0.25 ^a^	9.50 ± 1.63 ^a^	12.67 ± 1.12 ^ab^	11.52 ± 0.78 ^a^	12.19 ± 0.21 ^a^	12.33 ± 0.30 ^a^	12.33 ± 0.37 ^a^	11.11 ± 0.55	12.02 ± 0.61 ^a^	13.99 ± 1.45 ^ab^	12.90 ± 0.17 ^b^
**Gly**	7.78 ± 1.16 ^abc^	8.33 ± 0.21 ^a^	6.08 ± 0.62 ^bc^	5.57 ± 0.60	6.81 ± 0.40 ^c^	6.84 ± 0.19 ^ac^	7.74 ± 2.44 ^abc^	5.72 ± 0.23 ^b^	6.26 ± 0.41 ^ab^	6.63 ± 0.41 ^ab^	7.93 ± 1.06 ^ab^	8.02 ± 0.72 ^a^	7.36 ± 0.42 ^ab^	6.97 ± 0.32 ^b^
**Ala**	8.73 ± 1.02 ^ac^	8.96 ± 0.60 ^ac^	9.30 ± 1.03 ^ac^	7.96 ± 0.87 ^c^	11.43 ± 0.59	8.93 ± 0.82 ^abc^	10.08 ± 1.25 ^ac^	8.10 ± 0.31 ^bc^	7.76 ± 0.96 ^ab^	8.46 ± 0.21 ^ab^	9.13 ± 0.47 ^ab^	9.56 ± 0.30 ^b^	8.66 ± 0.44 ^b^	9.11 ± 0.72 ^b^
**Cys**	0.63 ± 0.06 ^a^	0.68 ± 0.06 ^a^	0.62 ± 0.06 ^a^	0.39 ± 0.03 ^a^	0.59 ± 0.30 ^ab^	0.58 ± 0.04	0.39 ± 0.03 ^a^	0.38 ± 0.01 ^a^	0.37 ± 0.00	0.46 ± 0.02	0.79 ± 0.28	0.93 ± 0.03 ^a^	0.68 ± 0.31 ^ab^	0.47 ± 0.03 ^b^
**Val**	4.67 ± 0.16 ^a^	4.86 ± 0.07 ^ab^	5.86 ± 0.16 ^a^	5.26 ± 0.69 ^b^	6.47 ± 0.44	5.29 ± 0.40 ^ab^	5.73 ± 0.18 ^a^	5.51 ± 0.30 ^ab^	4.78 ± 0.29 ^ab^	4.68 ± 0.02 ^a^	4.57 ± 0.60 ^ab^	4.71 ± 0.23 ^ab^	4.52 ± 0.26 ^ab^	5.04 ± 0.27 ^b^
**Met**	1.50 ± 0.47 ^ab^	1.72 ± 0.02 ^a^	1.95 ± 0.51 ^ab^	1.90 ± 0.08 ^a^	1.87 ± 0.23 ^a^	1.55 ± 0.04 ^ab^	1.54 ± 0.10 ^ab^	1.96 ± 0.18	2.32 ± 0.28 ^a^	2.20 ± 0.28 ^a^	2.40 ± 0.02 ^a^	1.56 ± 0.22 ^b^	2.10 ± 0.15	1.67 ± 0.09 ^b^
**Ile**	3.02 ± 0.18 ^ac^	3.20 ± 0.07 ^ac^	4.08 ± 0.46	3.92 ± 0.67 ^ac^	4.70 ± 0.44 ^a^	3.47 ± 0.25 ^ac^	3.71 ± 0.16 ^ab^	4.20 ± 0.45 ^b^	3.53 ± 0.15 ^a^	3.40 ± 0.14 ^a^	3.00 ± 0.50 ^ab^	2.75 ± 0.06 ^a^	3.00 ± 0.31 ^ab^	3.22 ± 0.05 ^b^
**Leu**	6.59 ± 0.45 ^ab^	6.71 ± 0.24 ^ab^	7.05 ± 0.44 ^ab^	6.89 ± 0.58 ^b^	8.15 ± 0.16	6.82 ± 0.18 ^ab^	6.92 ± 0.25 ^ab^	7.07 ± 0.14 ^a^	6.93 ± 0.18 ^a^	6.73 ± 0.30 ^ab^	6.30 ± 0.53 ^ab^	6.01 ± 0.16 ^a^	6.02 ± 0.27 ^a^	6.51 ± 0.10 ^b^
**Tyr**	6.85 ± 0.62 ^ab^	6.62 ± 0.18 ^ab^	5.71 ± 0.93 ^ab^	6.75 ± 0.87 ^b^	4.90 ± 0.71	6.90 ± 0.60 ^ab^	6.28 ± 1.35 ^ab^	6.90 ± 0.07 ^ab^	7.52 ± 0.01 ^a^	6.62 ± 0.55 ^bc^	7.19 ± 1.05 ^abc^	6.00 ± 0.18 ^a^	5.57 ± 0.55 ^a^	6.82 ± 0.30 ^c^
**Phe**	7.81 ± 5.30 ^ac^	5.19 ± 0.30 ^a^	5.96 ± 0.44 ^ac^	6.25 ± 0.90 ^ac^	5.70 ± 0.54 ^ac^	5.81 ± 0.07 ^ac^	5.81 ± 1.08 ^abc^	6.54 ± 0.18 ^b^	6.04 ± 0.31 ^ab^	5.68 ± 0.37 ^ab^	5.68 ± 0.58 ^ab^	5.27 ± 0.06 ^a^	5.10 ± 0.20 ^a^	5.67 ± 0.11 ^b^
**His**	3.93 ± 0.46 ^ab^	3.98 ± 0.36 ^ab^	3.70 ± 0.33 ^ab^	3.61 ± 0.15	3.25 ± 0.14	3.66 ± 0.10	3.18 ± 0.34 ^a^	3.32 ± 0.22 ^a^	3.54 ± 0.24 ^ab^	3.61 ± 0.56 ^ab^	3.43 ± 0.68 ^ab^	3.89 ± 0.07 ^ab^	2.82 ± 1.93 ^ab^	4.06 ± 0.33 ^b^
**Lys**	5.99 ± 0.84 ^a^	6.46 ± 0.26 ^a^	7.26 ± 1.91 ^ab^	6.62 ± 0.15	8.43 ± 1.35 ^b^	6.52 ± 0.33 ^a^	6.34 ± 0.32 ^a^	6.86 ± 0.68 ^ab^	7.18 ± 0.24 ^ab^	7.48 ± 0.59 ^ab^	6.94 ± 0.05 ^ab^	6.46 ± 0.16 ^a^	5.78 ± 1.78 ^ab^	7.05 ± 0.16 ^b^
**Arg**	4.42 ± 0.60 ^abc^	4.36 ± 0.15 ^ac^	3.72 ± 0.36 ^b^	5.03 ± 0.40 ^c^	2.71 ± 0.64	4.80 ± 0.21 ^abc^	4.22 ± 0.59 ^ac^	5.14 ± 0.33 ^bc^	5.29 ± 0.49 ^ab^	4.96 ± 0.25 ^ab^	4.42 ± 0.42 ^ab^	4.09 ± 0.28 ^ab^	5.74 ± 1.99 ^ab^	4.55 ± 0.34 ^b^
**OHPro**	0.39 ± 0.14 ^a^	0.27 ± 0.27 ^a^	3.03 ± 3.02 ^a^	3.78 ± 3.77 ^a^	0.62 ± 0.14 ^a^	0.00 ± 0.00	0.00 ± 0.00	0.00 ± 0.00	0.44 ± 0.08 ^a^	0.94 ± 0.56 ^a^	0.86 ± 0.48 ^a^	1.54 ± 0.79	4.05 ± 0.56	0.00 ± 0.00
**Pro**	5.35 ± 0.14 ^a^	5.18 ± 0.10 ^a^	7.03 ± 0.56	6.38 ± 0.87 ^ab^	7.80 ± 0.93 ^a^	6.31 ± 0.74 ^ab^	6.95 ± 0.45 ^a^	6.31 ± 0.28 ^a^	5.56 ± 0.52 ^ab^	5.91 ± 0.20 ^ab^	6.52 ± 0.76 ^ab^	6.45 ± 0.81 ^ab^	6.33 ± 0.31 ^a^	5.82 ± 0.17 ^b^
**TEAA/TAA**	41.39 ± 2.79 ^abc^	39.90 ± 0.69 ^a^	44.08 ± 2.72 ^bc^	43.82 ± 3.53 ^ac^	45.58 ± 2.24 ^a^	42.26 ± 1.68 ^ac^	41.98 ± 2.50 ^ac^	45.35 ± 1.39 ^a^	43.92 ± 1.32 ^a^	42.95 ± 1.10 ^ab^	40.49 ± 2.15 ^ab^	38.18 ± 0.51 ^a^	38.67 ± 2.73 ^ab^	41.97 ± 0.44 ^b^

Note: Values with the same letter display no significant differences by Tukey’s post hoc test for a significance of *p* < 0.05. Comparisons within different percentages are of the same diet.

**Table 7 insects-16-00113-t007:** Fatty acid content of black soldier fly larvae (% or g/100 g total fatty acids) supplied with different levels of replacement of tuna heads (TH: untreated tuna head, THH: tuna head hydrolysate, THO: tuna head oil) and codfish frames (CF: untreated cod frame, CFH: cod frame hydrolysate). *n*-3/*n*-6 is the ratio as a percentage between *n*-3 and *n*-6 fatty acids. Errors indicate the confidence intervals for *n* = 7 (replicates of processing) and α = 0.05.

Trials	TH100	TH50	TH25	THH50	THH25	THO10	THO5	THO2.5	CF100	CF50	CF25	CFH50	CFH25	C
**FA 12:0**	0.61 ± 0.14	3.93 ± 2.37 ^ab^	7.66 ± 5.03 ^ab^	12.79 ± 4.72 ^a^	14.90 ± 2.57 ^a^	10.17 ± 4.00 ^a^	4.69 ± 2.26 ^ab^	9.66 ± 3.67 ^a^	6.32 ± 2.07	18.77 ± 6.56 ^a^	25.58 ± 7.15 ^a^	4.70 ± 1.64 ^ab^	5.76 ± 2.68 ^ab^	2.45 ± 1.03 ^b^
**FA 14:0**	1.22 ± 0.18	4.26 ± 0.48	6.90 ± 0.96	8.38 ± 0.49	6.69 ± 0.28	6.51 ± 0.52 ^a^	6.09 ± 0.97 ^a^	8.69 ± 0.74	6.72 ± 1.04	11.38 ± 1.31 ^ab^	12.52 ± 2.05 ^ab^	8.12 ± 1.99 ^a^	5.61 ± 1.23 ^a^	12.47 ± 2.15 ^b^
**FA 15:0**	0.55 ± 0.04	0.67 ± 0.03	0.44 ± 0.04	0.32 ± 0.02	0.36 ± 0.01	0.47 ± 0.02	0.32 ± 0.02	0.25 ± 0.01	n.d.	n.d.	n.d.	n.d.	n.d.	n.d.
**FA 16:0**	25.30 ± 1.07 ^a^	23.51 ± 0.69 ^b^	23.64 ± 1.77 ^ab^	19.19 ± 0.99 ^a^	20.24 ± 0.53 ^a^	21.83 ± 1.52 ^a^	21.38 ± 0.46 ^a^	21.36 ± 0.99 ^a^	27.06 ± 4.83 ^a^	21.11 ± 2.96 ^a^	20.44 ± 4.68 ^a^	21.32 ± 3.28 ^a^	20.57 ± 1.47 ^a^	27.64 ± 3.37 ^c^
**FA 16:1**	10.29 ± 0.57	7.33 ± 0.45	4.87 ± 0.51	4.46 ± 0.33	6.43 ± 0.25	4.90 ± 0.22 ^a^	4.39 ± 0.35 ^a^	4.73 ± 0.23 ^a^	6.57 ± 1.44 ^a^	4.31 ± 0.77 ^b^	4.50 ± 1.24 ^abc^	4.39 ± 0.65	6.34 ± 0.99	3.40 ± 0.33 ^c^
**FA 17:0**	0.73 ± 0.06 ^a^	0.91 ± 0.02	0.74 ± 0.03 ^a^	0.56 ± 0.05 ^a^	0.53 ± 0.03 ^a^	0.72 ± 0.04	0.61 ± 0.05	0.50 ± 0.03	n.d.	n.d.	n.d.	n.d.	n.d.	n.d.
**FA 17:1**	1.31 ± 0.12	1.05 ± 0.04	0.48 ± 0.05	0.38 ± 0.04	0.43 ± 0.02	0.53 ± 0.02	0.44 ± 0.02	0.35 ± 0.03	n.d.	n.d.	n.d.	n.d.	n.d.	n.d.
**FA 18:0**	20.78 ± 2.28 ^c^	10.54 ± 1.76 ^a^	14.37 ± 3.71 ^ac^	6.13 ± 0.52 ^a^	7.31 ± 0.77 ^a^	9.84 ± 1.79 ^ab^	11.74 ± 1.02 ^a^	9.04 ± 0.66 ^b^	18.43 ± 6.89 ^ac^	12.38 ± 2.72 ^ac^	9.69 ± 5.91 ^ac^	10.59 ± 4.37 ^ac^	10.95 ± 3.07 ^ac^	18.11 ± 4.29 ^c^
**FA 18:1**	31.53 ± 2.47 ^a^	33.28 ± 1.08 ^a^	25.07 ± 2.16	21.65 ± 1.33 ^b^	28.91 ± 1.31	20.13 ± 0.81 ^b^	23.69 ± 1.08 ^ab^	22.52 ± 1.40 ^ab^	22.75 ± 7.21 ^ab^	19.44 ± 1.26 ^a^	17.05 ± 2.41 ^a^	23.51 ± 3.55 ^b^	31.10 ± 1.49	20.19 ± 2.59 ^b^
**FA 18:2 *n*-6**	2.82 ± 0.27	11.86 ± 0.21	13.58 ± 1.32	13.06 ± 0.70 ^a^	11.96 ± 0.40 ^a^	9.76 ± 0.25 ^a^	9.67 ± 0.35 ^a^	11.16 ± 0.54	4.78 ± 2.32 ^ac^	8.30 ± 1.23 ^bc^	7.58 ± 2.60 ^abc^	14.33 ± 2.71 ^a^	13.64 ± 0.52 ^a^	7.82 ± 1.08 ^c^
**FA 18:3 *n*-3**	n.d.	0.51 ± 0.02 ^ab^	0.60 ± 0.12 ^a^	0.85 ± 0.07	0.36 ± 0.03	0.72 ± 0.02 ^a^	0.58 ± 0.03	0.74 ± 0.05 ^a^	0.43 ± 0.06 ^ab^	0.38 ± 0.07 ^ab^	0.41 ± 0.09 ^ab^	5.24 ± 0.13	1.93 ± 0.06	0.42 ± 0.08 ^b^
**FA 20:4 *n-*6**	n.d.	n.d.	n.d.	0.96 ± 0.13	n.d.	1.14 ± 0.05 ^a^	1.13 ± 0.12 ^a^	0.72 ± 0.06	n.d.	n.d.	n.d.	0.37 ± 0.11	n.d.	n.d.
**FA 20:5 *n*-3**	0.74 ± 0.15 ^a^	1.11 ± 0.13	0.66 ± 0.08 ^a^	5.84 ± 0.68	1.18 ± 0.16	6.61 ± 0.21 ^a^	6.73 ± 0.62 ^a^	4.81 ± 0.32	4.71 ± 1.64 ^a^	2.88 ± 0.34 ^a^	1.60 ± 0.51	2.87 ± 0.63 ^a^	3.32 ± 0.17 ^a^	n.d.
**FA 22:2**	1.17 ± 0.57	0.47 ± 0.11 ^a^	0.30 ± 0.22 ^a^	0.71 ± 0.06	n.d.	0.67 ± 0.08	1.43 ± 0.10 ^a^	1.55 ± 0.13 ^a^	n.d.	n.d.	n.d.	n.d.	n.d.	n.d.
**FA 22:6 *n*-3**	2.94 ± 2.78 ^a^	0.57 ± 0.11 ^a^	0.68 ± 0.34 ^ab^	4.81 ± 0.63	0.69 ± 0.20	6.01 ± 0.31	7.10 ± 0.54	3.91 ± 0.26	2.10 ± 0.38	1.05 ± 0.11	0.63 ± 0.17	0.99 ± 0.28 ^a^	0.79 ± 0.17 ^a^	0.30 ± 0.04 ^b^
**∑** **SFA ^1^**	49.20 ± 2.40 ^a^	43.82 ± 1.25	53.75 ± 3.54 ^a^	47.28 ± 3.74 ^a^	50.03 ± 1.92 ^a^	49.54 ± 1.61 ^a^	44.84 ± 2.38 ^b^	49.50 ± 2.60 ^ab^	57.91 ± 12.14 ^ac^	63.65 ± 3.24 ^ac^	68.23 ± 5.67 ^ac^	48.30 ± 8.46 ^a^	42.88 ± 1.65 ^a^	66.46 ± 4.04 ^c^
**∑** **MUFAs ^2^**	43.13 ± 2.84 ^a^	41.65 ± 1.09 ^a^	30.42 ± 2.20	26.49 ± 1.60 ^c^	35.77 ± 1.40	25.56 ± 0.85 ^ac^	28.52 ± 1.14 ^b^	27.60 ± 1.47 ^ab^	29.32 ± 8.58 ^a^	23.75 ± 1.78 ^ac^	21.55 ± 3.49 ^ac^	27.90 ± 4.07	37.44 ± 1.92	23.58 ± 2.91 ^c^
**∑** **PUFAs ^3^**	7.67 ± 2.36 ^c^	14.52 ± 0.27 ^a^	15.83 ± 1.44 ^a^	26.23 ± 2.24	14.19 ± 0.70	24.90 ± 0.83 ^ab^	26.63 ± 1.48 ^a^	22.90 ± 1.21 ^b^	12.78 ± 4.77 ^ac^	12.60 ± 1.53 ^ac^	10.22 ± 3.12 ^ac^	23.80 ± 4.59 ^a^	19.68 ± 1.04 ^a^	9.96 ± 1.50 ^c^
**∑** ** *n* ** **-3**	3.67 ± 2.51 ^ab^	2.19 ± 0.19 ^ab^	1.95 ± 0.27 ^ab^	11.50 ± 1.37	2.23 ± 0.35 ^b^	13.33 ± 0.51 ^a^	14.41 ± 1.08 ^a^	9.46 ± 0.57	8.00 ± 3.62	4.31 ± 0.44	2.65 ± 0.55 ^b^	8.49 ± 1.88	6.04 ± 0.56	0.72 ± 0.44 ^b^
**∑** ** *n* ** **-6**	2.82 ± 0.25	11.86 ± 0.21	13.85 ± 1.23	14.02 ± 0.70 ^a^	11.96 ± 0.40 ^a^	10.90 ± 0.25 ^a^	10.80 ± 0.33 ^a^	11.88 ± 0.50	4.78 ± 2.32 ^ab^	8.30 ± 1.23 ^ab^	7.58 ± 2.60 ^ab^	1.34 ± 0.19	13.64 ± 0.52	7.82 ± 1.08 ^b^
** *n* ** **-3/*n*-6**	1.32 ± 0.93	0.19 ± 0.02	0.14 ± 0.02	0.82 ± 0.06	0.19 ± 0.02	1.22 ± 0.03	1.33 ± 0.08	0.80 ± 0.02	2.02 ± 1.38	0.53 ± 0.06	0.36 ± 0.06	6.33 ± 0.02	0.44 ± 0.03	0.09 ± 0.02

Note: Values with the same letter display no significant differences by Tukey’s post hoc test for a significance of *p* < 0.05. Comparisons within different percentages of the same diet. ^1^ Saturated fatty acids. ^2^ Monounsaturated fatty acids. ^3^ Polyunsaturated fatty acids.

## Data Availability

Data provided in Appendix A.

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
