# Peer review of "Growth Rate Prediction, Performance, and Biochemical Enhancement of Black Soldier Fly (Hermetia illucens) Fed with Marine By-Products and Co-Products: A Potential Value-Added Resource for Marine Aquafeeds"

_insects, 2025, doi:10.3390/insects16020113_

Round 1
Reviewer 1 Report
Comments and Suggestions for Authors
The present paper tested the potentiality of marine derived by-products on BSF survival, growth and quality.
In general the paper is not that new since many other papers have already tested similar experiments showing that PUFAs content in BSF larvae can be increased up to a certain level by changing the diet but that other FAs such as C12 remain high.
Also the viscosity and moisture of the diets have been tested and these results have already been identified.
The MS is well written even though in some parts it is a little speculative. My major concern is that the authors replaced from 0 to 100% of the control diets with the sustainable ingredients but it would have been better to formulate the new diets in order to be isoenerhetic-lipidic and proteic respect to the control. How can you compare the results if you did not took into account this aspect?
specific comments:
lines 54-61 most of the present aquafeeds are largely vegetable and this should pointed out.
Please latin names should be in italics
112-114 please provide details about mincing and liophilization. Brand of equipment, size of samples, total amount produced.
l145 why is this control diet? is this a comemrcial diet for insects?Any reference?
Humidity is low usually it is around 70 % as furthter stated in the MS
Weightening: please indicate intsrument used and precision
l168: how did you recognize the prepupae stage?
305-307 please try to find some papers with similar results to strenght your results
312-319 why you did not adjust moisture with distilled water?
Diet analysis is missing Energy content which can be very important. please add.
523-536 the authors missed different published papers on completely vegetable diets including low amounts of insect meal in trout seabream and seabass which can be find easily
581 please explain the role of C12 in fish
609-614 not really new
Authors should also consider the role and amount of chitin
Author Response
Reviewer #1. General comments
The present paper tested the potentiality of marine derived by-products on BSF survival, growth and quality.
In general the paper is not that new since many other papers have already tested similar experiments showing that PUFAs content in BSF larvae can be increased up to a certain level by changing the diet but that other FAs such as C12 remain high.
Also the viscosity and moisture of the diets have been tested and these results have already been identified.
The MS is well written even though in some parts it is a little speculative. My major concern is that the authors replaced from 0 to 100% of the control diets with the sustainable ingredients but it would have been better to formulate the new diets in order to be isoenerhetic-lipidic and proteic respect to the control. How can you compare the results if you did not took into account this aspect?
Reply: Indeed, as well remarked by the reviewer, the diets used in this study were not formulated to be isoenergetic, isoproteic, or isolipidic relative to the control diet. However, to the best of our knowledge, this does not compromise the validity of our findings, as similar studies are currently available in the scientific literature addressing this topic without this standardization being applied. One must highlight that these are practical diets and that all feeds were provided at a rate of 15 mg per larva per day, thus allowing larvae to have an unrestricted access to food (ad libitum feeding), meaning that daily energy intake was not controlled.
The primary objective of this study is to determine, under consistent husbandry conditions and with practical diets differing in protein and lipid content, the threshold levels of essential nutrient incorporation (amino acids and fatty acids) required for larval development and marine aquafeeds. Additionally, we aimed to explore whether enhancing the bioavailability of these essential nutrients would result in a different threshold for their incorporation in BSF.
specific comments
Reviewer #1: lines 54-61 most of the present aquafeeds are largely vegetable and this should pointed out.
Reply: In lines 54-61 we have specifically pointed out what the reviewer mentioned:
” However, several questions are raised due to the amount of feed needed to supply the animal species commonly produced in this industry, namely because a significant part of the ingredients used in aquafeeds still promote environmental constraints [4]. Fish meal and fish oil, soybeans, oilseeds, vegetable meals, etc. are some of the ingredients most commonly employed in the formulation of aquafeeds [5,6]. The environmental footprint of each ingredient is a reality that cannot be neglected anymore and has gained much interest in recent years”
Reviewer #1: Please latin names should be in italics
Reply: Corrected as recommended throughout the revised manuscript.
Reviewer #1: 112-114 please provide details about mincing and liophilization. Brand of equipment, size of samples, total amount produced.
Reply: The requested details have now been incorporated in the revised manuscript. The text now reads as follows: “All biological material was minced, up to size of 20-40 x 20-40 mm using a meat mincer (Mobba PM32, Mobba Industrial Catalunya, Badalona, Spain) and stored at -18 °C until further use. These raw materials were then freeze-dried (LyoBeta, Syntegon Telstar SLU, Terrassa, Spain) to obtain crude substrates that were studied as potential feed ingredients (TH and CF) for BSF larvae. The amount of processed materials was around 20 kg of each by-product.” (Line 112-117)
Reviewer #1: l145 why is this control diet? is this a comemrcial diet for insects? Any reference?
Reply: As noted in line 331, "Moreover, chicken feed is commonly used as a control, as this is known to be a diet under which BSF can thrive with good results [53]." Chicken feed is frequently employed as a control diet, not only in the present study but also in numerous others (e.g., Liland et al., 2017; Oonincx et al., 2015; Tinder et al., 2017). This diet, formulated commercially for chicks rather than insects, serves as a standardized reference, and its specific details have been provided in the manuscript; the text now reads as follows: “A control diet consisting of chick feed (Zêzere A-104 Pintos Migalha: crude protein 17%, crude lipid 3.5%, ash 5% and crude fibre 5.5%) and tap water ...” (Lines 147-148)
- Liland, N.S., Biancarosa, I., Araujo, P., Biemans, D., Bruckner, C.G., Waagbø, R., Torstensen, B.E., Lock, E.-J.J., 2017. Modulation of nutrient composition of black soldier fly (Hermetia illucens) larvae by feeding seaweed-enriched media. PLoS One 12, e0183188. https://doi.org/10.1371/journal.pone.0183188
- Oonincx, D., Huis, A., van Loon, J., 2015. Nutrient utilisation by black soldier flies fed with chicken, pig, or cow manure, Journal of Insects as Food and Feed. https://doi.org/10.3920/JIFF2014.0023
- Tinder, A.C., Puckett, R.T., Turner, N.D., Cammack, J.A., Tomberlin, J., 2017. Bioconversion of sorghum and cowpea by black soldier fly (Hermetia illucens (L.)) larvae for alternative protein production. J. Insects as Food Feed 3, 1–10. https://doi.org/10.3920/JIFF2016.0048
Reviewer #1: Humidity is low usually it is around 70 % as furthter stated in the MS
Reply: Yes, we confirm this statement.
Reviewer #1: Weightening: please indicate intsrument used and precision
Reply: The requested details have now been incorporated into the manuscript. The text now reads as follows: “Each group, with 50 individuals, was then weighed on an electronic weighting scale (using a RADWAG AS 220.R2 analytical balance) to determine the collective average weight and consequently...” (Line 153-155);” Weight gain was controlled by weighing a random sample of 10 larvae during the trials (using a RADWAG AS 220.R2 analytical balance) and returning adding them to their container of origin again....” (Line 170)
Reviewer #1: l168: how did you recognize the prepupae stage?
Reply: At the onset of the prepupal stage, larvae reduce feeding activity and begin to develop darker pigmentation around their body segments. This darkening continues progressively until feeding ceases entirely, at which point the larvae become fully darkened and initiate migration away from the feeding substrate. During this phase, the developing larvae transition into the pupal stage. For clarity, this information was now added to the revised manuscript and it now reads as follows: “On the 10th day (the 17th day of the life cycle), the larvae were harvested, already at the pre-pupa stage (when larvae reduce their feeding activity and begin to develop darker pigmentation around their body segments and initiate migration away from the feeding substrate; this pupal stage the most used BSF life stage in insect meal production), and then starved”. (Line 173-176)
Reviewer #1: 305-307 please try to find some papers with similar results to strenght your results
Reply: The requested details have now been incorporated into the revised manuscript. The text now reads as follows:”. A recent study by Kießling et al. [47] observed that substrates with high lipid content (>15%) led to excessive free fat accumulation, creating anaerobic and adhe-sive conditions on the larval surface. These conditions impair larval respiration and hin-der growth. Based on these findings, those authors concluded that a crude lipid content of approximately 10% on a dry matter basis is optimal for achieving an efficient weight gain and promoting efficient larval performance [47]. In fact, oils are used as pesticides mostly killing insects on contact by disrupting gas exchange (respiration) [48].” (Line 317-324)
- [47] M. Kieβling, K. Franke, V. Heinz, and K. Aganovic, Relationship between substrate composition and larval weight: a simple growth model for black soldier fly larvae, Journal of Insects as Food and Feed 2023 9:8, 1027-1036 https://doi.org/10.3920/JIFF2022.0096open_in_new
- [48] Seni, A. Potential of the various oils for insect pests’ management and their effect on beneficial insects. Int. J. Trop. Insect Sci. 2023, 43, 321–337.
Reviewer #1: 312-319 why you did not adjust moisture with distilled water?
Reply: We apologise for the misleading usage of the term moisture; we were referring to the free fat content of diets, not their moisture. As such, this issue was not associated with water level adjustments in the feed but rather with the presence of excessive fat, which adversely affected larval performance and viability. We have corrected this mistake in our revised manuscript and the text now reads as follows:” It is consensual that both growth and the nutritional profile of insects are influenced by the diet supplied. No BSFL survived when fed THH100 or CFH100; consequently, these trials are not further discussed in this study. In the case of THH 100, the amount of fat in the diet was such that it created a thick film on the top of this feeding substrate that led the larvae to drown. A recent study by Kießling et al. [47] observed that substrates with high lipid content (>15%) led to excessive free fat accumulation, creating anaerobic and adhe-sive conditions on the larval surface. These conditions impair larval respiration and hin-der growth. Based on these findings, those authors concluded that a crude lipid content of approximately 10% on a dry matter basis is optimal for achieving an efficient weight gain and promoting efficient larval performance [47]. In fact, oils are used as pesticides mostly killing insects on contact by disrupting gas exchange (respiration) [48].
For the CFH100 trials, the viscosity was so high that the larvae were not even able to move, which ultimately caused their death as well. For the remaining feeding trials, the best survival was recorded when supplying BSFL with CF100 and CFH25 (both reaching 99.7%), though in general, feeding trials employing CF displayed a survival always above 98% (Table 1). For TH, the lowest recorded survival was that of THH50 (30.9%), while the best result in terms of survival was achieved using THO (95.7%). These results suggest that the fat content of the diets employed could be one of the parameters most influencing survival, as already documented by other author studying the culture of BSFL under different diets [47]. Significant larval growth can still occur at low protein concentrations (e.g., 5%) when the crude lipid content is relatively high (e.g., approximately 12%). How-ever, even slight variations in lipid content, such as an increase from 5% to 12%, appear to lead to noticeable reductions in weight gain [47]. These findings underscore the im-portance of maintaining optimal substrate properties for BSF cultivation, as inappropriate diet formulations can negatively impact larval survival, growth efficiency, and overall yield. This highlights the need for careful consideration of diet composition in large-scale BSF production systems to ensure both productivity and animal welfare.“ (Line 313-341)
Reviewer #1: Diet analysis is missing Energy content which can be very important. please add.
Reply: As requested by the reviewer, a column with the energy content of each diet provided to BSF was now added to Table 5 in our revised manuscript. To better accommodate the inclusion of this new data the following text was added for clarity: “Energy estimation was calculated according to Codex Alimentarius [43] using the general factors for energy conversion: 17 kJ/g (4 kcal/g), 37 kJ/g (9 kcal/g) and 17 kJ/g (4 kcal/g), for protein, fat and carbohydrate, respectively.” (Line 219-222)
- [43] Alimentarius, C. Joint FAO/WHO Food Standards Programme Codex Alimentarius Commission, Rome, 1991.
Reviewer #1: 523-536 the authors missed different published papers on completely vegetable diets including low amounts of insect meal in trout seabream and seabass which can be find easily
Reply: The requested information has been incorporated into the manuscript. However, we have selected marine species as examples instead of freshwater species, as they are more aligned with the primary objective of this study: enhancing the n-3 PUFA content in BSF larvae. The text now reads as follows:” These findings highlight the remarkable nutritional potential of BSF as a protein source, further supported by its demonstrated suitability as an alternative feed ingredient for aq-uaculture diets, with no observed negative impacts on growth performance, or overall health such as white Pacific shrimp (Litopeneaus vannamei), rainbow trout (Oncorhynchus mykiss) or Atlantic salmon (Salmo salar). [74-77].”. (Line 566-571)
- [74] Cummins, V.C.; Rawles, S.D.; Thompson, K.R.; Velasquez, A.; Kobayashi, Y.; Hager, J.; Webster, C.D. Evaluation of Black Soldier Fly (Hermetia illucens) larvae meal as partial or total replacement of marine fish meal in practical diets for Pacific white shrimp (Litopenaeus vannamei). Aquaculture 2017, 473, 337–344.
- [75] Dumas, A.; Raggi, T.; Barkhouse, J.; Lewis, E.; Weltzien, E. The oil fraction and partially defatted meal of black soldier fly larvae (Hermetia illucens) affect differently growth performance, feed efficiency, nutrient deposition, blood glucose and lipid digestibility of rainbow trout (Oncorhynchus mykiss). Aquaculture 2018, 492, 24–34.
- [76] Lock, E.R.; Arsiwalla, T.; Waagbø, R. Insect larvae meal as an alternative source of nutrients in the diet of Atlantic salmon (Salmo salar) Postsmolt. Aquac. Nutr. 2016, 22, 1202–1213.
- [77] Renna, M.; Schiavone, A.; Gai, F.; Dabbou, S.; Lussiana, C.; Malfatto, V.; Prearo, M.; Capucchio, M.T.; Biasato, I.; Biasibetti, E.; et al. Evaluation of the suitability of a partially defatted black soldier fly (Hermetia illucens,) larvae meal as ingredient for rainbow trout (Oncorhynchus mykiss, Walbaum) Diets. J. Anim. Sci. Biotechnol. 2017, 8, 57.
Reviewer #1: 581 please explain the role of C12 in fish
Reply: This information was added to the manuscript. The text now reads as follows:” Among the SFA, the most abundant was the FA 16:0 (palmitoleic acid) (19.2-27.6%) followed by the FA 18:0 (stearic acid) (6.1-20.8%). Notably, the FA 12:0 (lauric acid), recognized for its positive effect in growth performance and FA balance in Nile tilapia [79] and Atlantic salmon [80],...” (Line 610-613)
- [79] Ashraf M.A. Goda, Ehab El-Haroun, Hani Nazmi, Hien Van Doan, Ahmed M. Aboseif, Mostafa K.S. Taha, Nevine.M. Abou Shabana, Black soldier fly oil-based diets enriched in lauric acid enhance growth, hematological indices, and fatty acid profiles of Nile tilapia, Oreochromis niloticus fry, Aquaculture Reports, 37, 2024, https://doi.org/10.1016/j.aqrep.2024.102269.
- [80] Belghit I, Waagbø R, Lock E-J, Liland NS. Insect-based diets high in lauric acid reduce liver lipids in freshwater Atlantic salmon. Aquacult Nutr. 2019; 25: 343–357. https://doi.org/10.1111/anu.1286
Reviewer #1: 609-614 not really new
Reply: To the best of the authors knowledge, no previous studies have performed feeding trials using hydrolysed feeds to enhance the lipid profile of BSF larvae. Consequently, we consider that the present findings represent novel contributions, that advance the state of the art on this topic. Additionally, the high levels of n-3 fatty acids bioaccumulated in BSF larval tissue observed in this study are rarely reported in other studies that try to address similar objectives.
Reviewer #1: Authors should also consider the role and amount of chitin
Reply: While this is an important topic, we have not determined chitin levels in the present study and, therefore, any discussion on this topic would rather be speculative. The primary objective of this work was to evaluate the threshold levels of essential fatty acids and amino acids that BSF can incorporate under enhanced conditions of bioavailability to improve its biochemical profile for applications in aquafeed formulation for marine species. The authors will certainly keep the recommendation of the reviewer in mind for future studies on this topic, as the prevalence of chitin in the biomass of BSF destined to be used as an ingredient in aquafeed formulation is certainly worth addressing.

Reviewer 2 Report
Comments and Suggestions for Authors
1.) Statistical analysis and how results were presented can be improved:
*Determine linear contrasts for each pretreatment. This is to answer what is likely the optimal inclusion rate for each pretreatment or oil extraction.
*Determine impact of different pretreatments on BSFL growth and nutrient composition. What is the ideal method of pretreatment for CF and TH?
2.) Discussion of results could be improved:
*What was the reason why untreated/lyophilization appear to be better for BSFL growth than hydrolysis?
3.) Add implications
Author Response
Reviewer #2.
1.) Statistical analysis and how results were presented can be improved:
*Determine linear contrasts for each pretreatment. This is to answer what is likely the optimal inclusion rate for each pretreatment or oil extraction.
Reply: Following the reviewer recommendation, we have included the linear correlations for each treatment; please refer to Supporting Information (Figure S1).
Reviewer #2. *Determine impact of different pretreatments on BSFL growth and nutrient composition. What is the ideal method of pretreatment for CF and TH?
Reply: This information was inserted in the revised text of our manuscript, and it now reads as follows: “To select the best substrate candidate, we calculated the simple linear correlation be-tween growth responses (survival, bioconversion efficiency and maximum growth-Xm) and levels of replacement (Figure S1, supporting information). For TH, the level of 50% is the compromise option, resulting in the highest survival, bioconversion and interesting biomass production (slightly less in THH100). For THH and CFH, only two levels could not be linearized, and the lowest percentages directly showed the best performances (25% in both cases). In this context, the lowest incorporations of tuna oil and cod frame by-product (THO2.5 and CF25) also proven to be the most adequate substrates for BSF growth. Based on these results, the best options in terms of maximum values of the mentioned growth variables were THO2.5 and THH25. “(Line 447-456)
Reviewer #2. 2.) Discussion of results could be improved:
*What was the reason why untreated/lyophilization appear to be better for BSFL growth than hydrolysis?
Reply: The untreated diet provided a texture that allowed BSF larvae to move freely and exhibit natural behaviours. In contrast, the addition of hydrolysates significantly altered the texture of diets, increasing viscosity and forming a dense fat layer on its surface. This modification impaired substrate aeration and restricted larval mobility, ultimately leading to a decrease in their growth and, in their upper levels of replacement, to drowning and suffocation. This information has been highlighted in the manuscript and now reads as follows: “, the amount of fat in the diet was such that it created a thick film on the top of this feeding substrate that led the larvae to drown. A recent study by Kießling et al. [47] observed that substrates with high lipid content (>15%) led to excessive free fat accumulation, creating anaerobic and adhesive conditions on the larval surface. These conditions impair larval respiration and hinder growth. Based on these findings, those authors concluded that a crude lipid content of approximately 10% on a dry matter basis is optimal for achieving an efficient weight gain and promoting efficient larval performance [47]. In fact, oils are used as pesticides mostly killing insects on contact by disrupting gas exchange (respiration) [48].
For the CFH100 trials, the viscosity was so high that the larvae were not even able to move, which ultimately caused their death as well. For the remaining feeding trials, the best survival was recorded when supplying BSFL with CF100 and CFH25 (both reaching 99.7%), though in general, feeding trials employing CF displayed a survival always above 98% (Table 1). For TH, the lowest recorded survival was that of THH50 (30.9%), while the best result in terms of survival was achieved using THO (95.7%). These results suggest that the fat content of the diets employed could be one of the parameters most influencing survival, as already documented by other author studying the culture of BSFL under different diets [47]. Significant larval growth can still occur at low protein concentrations (e.g., 5%) when the crude lipid content is relatively high (e.g., approximately 12%)....” (Line 315-335)
Reviewer #2. 3.) Add implications
Reply: To best accommodate this recommendation by the reviewer, we have added the following information to our revised text: “These findings underscore the importance of maintaining optimal substrate properties for BSF cultivation, as inappropriate diet formulations can negatively impact larval survival, growth efficiency, and overall yield. This highlights the need for careful consideration of diet composition in large-scale BSF production systems to ensure both productivity and animal welfare.” (Line 336-341).

Round 2
Reviewer 1 Report
Comments and Suggestions for Authors
Mant thanks for undwering most of the comments. However i still think that some issues have to be addressed befor the MS publication.
Reviewer #1. General comments
My major concern is that the authors replaced from 0 to 100% of the control diets with the sustainable ingredients but it would have been better to formulate the new diets in order to be isoenerhetic-lipidic and proteic respect to the control. How can you compare the results if you did not took into account this aspect?
Reply: Indeed, as well remarked by the reviewer, the diets used in this study were not formulated to be isoenergetic, isoproteic, or isolipidic relative to the control diet. However, to the best of our knowledge, this does not compromise the validity of our findings, as similar studies are currently available in the scientific literature addressing this topic without this standardization being applied. One must highlight that these are practical diets and that all feeds were provided at a rate of 15 mg per larva per day, thus allowing larvae to have an unrestricted access to food (ad libitum feeding), meaning that daily energy intake was not controlled.
I agree with this but then the autors should also think to show the remaining feed in the MS so that the reader can better understand which was in reality the insect feed intake.
The primary objective of this study is to determine, under consistent husbandry conditions and with practical diets differing in protein and lipid content, the threshold levels of essential nutrient incorporation (amino acids and fatty acids) required for larval development and marine aquafeeds. Additionally, we aimed to explore whether enhancing the bioavailability of these essential nutrients would result in a different threshold for their incorporation in BSF.
Ok , but why only marine aquafeed? This is not clear from the title. Growth rate prediction, performance, and biochemical enhancement of Black Soldier Fly (Hermetia illucens) fed with marine by-products and co-products: a potential value-added resource for aquafeeds
specific comments
Reviewer #1: lines 54-61 most of the present aquafeeds are largely vegetable and this should pointed out.
Reply: In lines 54-61 we have specifically pointed out what the reviewer mentioned:
” However, several questions are raised due to the amount of feed needed to supply the animal species commonly produced in this industry, namely because a significant part of the ingredients used in aquafeeds still promote environmental constraints [4]. Fish meal and fish oil, soybeans, oilseeds, vegetable meals, etc. are some of the ingredients most commonly employed in the formulation of aquafeeds [5,6]. The environmental footprint of each ingredient is a reality that cannot be neglected anymore and has gained much interest in recent years”
The text is very general, and may be it is better to elucidate that nowadays most of the diets are largely vegetable.
Reviewer #1: l145 why is this control diet? is this a comemrcial diet for insects? Any reference?
Reply: As noted in line 331, "Moreover, chicken feed is commonly used as a control, as this is known to be a diet under which BSF can thrive with good results [53]." Chicken feed is frequently employed as a control diet, not only in the present study but also in numerous others (e.g., Liland et al., 2017; Oonincx et al., 2015; Tinder et al., 2017). This diet, formulated commercially for chicks rather than insects, serves as a standardized reference, and its specific details have been provided in the manuscript; the text now reads as follows: “A control diet consisting of chick feed (Zêzere A-104 Pintos Migalha: crude protein 17%, crude lipid 3.5%, ash 5% and crude fibre 5.5%) and tap water ...” (Lines 147-148)
- Liland, N.S., Biancarosa, I., Araujo, P., Biemans, D., Bruckner, C.G., Waagbø, R., Torstensen, B.E., Lock, E.-J.J., 2017. Modulation of nutrient composition of black soldier fly (Hermetia illucens) larvae by feeding seaweed-enriched media. PLoS One 12, e0183188. https://doi.org/10.1371/journal.pone.0183188
- Oonincx, D., Huis, A., van Loon, J., 2015. Nutrient utilisation by black soldier flies fed with chicken, pig, or cow manure, Journal of Insects as Food and Feed. https://doi.org/10.3920/JIFF2014.0023
- Tinder, A.C., Puckett, R.T., Turner, N.D., Cammack, J.A., Tomberlin, J., 2017. Bioconversion of sorghum and cowpea by black soldier fly (Hermetia illucens (L.)) larvae for alternative protein production. J. Insects as Food Feed 3, 1–10. https://doi.org/10.3920/JIFF2016.0048
yes that can be an answer, but there are many other diets that are used as control diet for HI. May be better specify that in the present study in reference to ...the followong fiets was used as reference diet.
Reviewer #1: 305-307 please try to find some papers with similar results to strenght your results
Reply: The requested details have now been incorporated into the revised manuscript. The text now reads as follows:”. A recent study by Kießling et al. [47] observed that substrates with high lipid content (>15%) led to excessive free fat accumulation, creating anaerobic and adhe-sive conditions on the larval surface. These conditions impair larval respiration and hin-der growth. Based on these findings, those authors concluded that a crude lipid content of approximately 10% on a dry matter basis is optimal for achieving an efficient weight gain and promoting efficient larval performance [47]. In fact, oils are used as pesticides mostly killing insects on contact by disrupting gas exchange (respiration) [48].” (Line 317-324)
- [47] M. Kieβling, K. Franke, V. Heinz, and K. Aganovic, Relationship between substrate composition and larval weight: a simple growth model for black soldier fly larvae, Journal of Insects as Food and Feed 2023 9:8, 1027-1036 https://doi.org/10.3920/JIFF2022.0096open_in_new
- [48] Seni, A. Potential of the various oils for insect pests’ management and their effect on beneficial insects. Int. J. Trop. Insect Sci. 2023, 43, 321–337.
Thanks, also consider similar papers that increased the PUFAs level in HI
Reviewer #1: Diet analysis is missing Energy content which can be very important. please add.
Reply: As requested by the reviewer, a column with the energy content of each diet provided to BSF was now added to Table 5 in our revised manuscript. To better accommodate the inclusion of this new data the following text was added for clarity: “Energy estimation was calculated according to Codex Alimentarius [43] using the general factors for energy conversion: 17 kJ/g (4 kcal/g), 37 kJ/g (9 kcal/g) and 17 kJ/g (4 kcal/g), for protein, fat and carbohydrate, respectively.” (Line 219-222)
- [43] Alimentarius, C. Joint FAO/WHO Food Standards Programme Codex Alimentarius Commission, Rome, 1991.
Reviewer #1: 523-536 the authors missed different published papers on completely vegetable diets including low amounts of insect meal in trout seabream and seabass which can be find easily
Reply: The requested information has been incorporated into the manuscript. However, we have selected marine species as examples instead of freshwater species, as they are more aligned with the primary objective of this study: enhancing the n-3 PUFA content in BSF larvae. The text now reads as follows:” These findings highlight the remarkable nutritional potential of BSF as a protein source, further supported by its demonstrated suitability as an alternative feed ingredient for aq-uaculture diets, with no observed negative impacts on growth performance, or overall health such as white Pacific shrimp (Litopeneaus vannamei), rainbow trout (Oncorhynchus mykiss) or Atlantic salmon (Salmo salar). [74-77].”. (Line 566-571)
- [74] Cummins, V.C.; Rawles, S.D.; Thompson, K.R.; Velasquez, A.; Kobayashi, Y.; Hager, J.; Webster, C.D. Evaluation of Black Soldier Fly (Hermetia illucens) larvae meal as partial or total replacement of marine fish meal in practical diets for Pacific white shrimp (Litopenaeus vannamei). Aquaculture 2017, 473, 337–344.
- [75] Dumas, A.; Raggi, T.; Barkhouse, J.; Lewis, E.; Weltzien, E. The oil fraction and partially defatted meal of black soldier fly larvae (Hermetia illucens) affect differently growth performance, feed efficiency, nutrient deposition, blood glucose and lipid digestibility of rainbow trout (Oncorhynchus mykiss). Aquaculture 2018, 492, 24–34.
- [76] Lock, E.R.; Arsiwalla, T.; Waagbø, R. Insect larvae meal as an alternative source of nutrients in the diet of Atlantic salmon (Salmo salar) Postsmolt. Aquac. Nutr. 2016, 22, 1202–1213.
- [77] Renna, M.; Schiavone, A.; Gai, F.; Dabbou, S.; Lussiana, C.; Malfatto, V.; Prearo, M.; Capucchio, M.T.; Biasato, I.; Biasibetti, E.; et al. Evaluation of the suitability of a partially defatted black soldier fly (Hermetia illucens,) larvae meal as ingredient for rainbow trout (Oncorhynchus mykiss, Walbaum) Diets. J. Anim. Sci. Biotechnol. 2017, 8, 57.
These papers are not very up do date. Additionally, some papers about mediterranena species should be of interest.
Reviewer #1: 581 please explain the role of C12 in fish
Reply: This information was added to the manuscript. The text now reads as follows:” Among the SFA, the most abundant was the FA 16:0 (palmitoleic acid) (19.2-27.6%) followed by the FA 18:0 (stearic acid) (6.1-20.8%). Notably, the FA 12:0 (lauric acid), recognized for its positive effect in growth performance and FA balance in Nile tilapia [79] and Atlantic salmon [80],...” (Line 610-613)
This is actually not always true, especially if, as you stated previously, you are focusing on marine species. C12 is important in avoiding intestinal inflammation in fish fed on HI diets. For this reason the next comment of chitin is importanty and sgould at least be mentioned in the discussion. Please revise
- [79] Ashraf M.A. Goda, Ehab El-Haroun, Hani Nazmi, Hien Van Doan, Ahmed M. Aboseif, Mostafa K.S. Taha, Nevine.M. Abou Shabana, Black soldier fly oil-based diets enriched in lauric acid enhance growth, hematological indices, and fatty acid profiles of Nile tilapia, Oreochromis niloticus fry, Aquaculture Reports, 37, 2024, https://doi.org/10.1016/j.aqrep.2024.102269.
- [80] Belghit I, Waagbø R, Lock E-J, Liland NS. Insect-based diets high in lauric acid reduce liver lipids in freshwater Atlantic salmon. Aquacult Nutr. 2019; 25: 343–357. https://doi.org/10.1111/anu.1286
Reviewer #1: 609-614 not really new
Reply: To the best of the authors knowledge, no previous studies have performed feeding trials using hydrolysed feeds to enhance the lipid profile of BSF larvae. Consequently, we consider that the present findings represent novel contributions, that advance the state of the art on this topic. Additionally, the high levels of n-3 fatty acids bioaccumulated in BSF larval tissue observed in this study are rarely reported in other studies that try to address similar objectives.
I agree but please be sure to correctly cite all the papers that improved the PUFAs content in HI through different approaches
Reviewer #1: Authors should also consider the role and amount of chitin
Reply: While this is an important topic, we have not determined chitin levels in the present study and, therefore, any discussion on this topic would rather be speculative. The primary objective of this work was to evaluate the threshold levels of essential fatty acids and amino acids that BSF can incorporate under enhanced conditions of bioavailability to improve its biochemical profile for applications in aquafeed formulation for marine species. The authors will certainly keep the recommendation of the reviewer in mind for future studies on this topic, as the prevalence of chitin in the biomass of BSF destined to be used as an ingredient in aquafeed formulation is certainly worth addressing.
yes and there are already many published papers about the role of chitn. So i kindly suggest to add it to the text at least as a comment.
Author Response
Reply to the reviewer comments
Dear Editor, we start by acknowledging the positive criticism of anonymous reviewer #1, we have tried to address, in the best way possible, all the issues raised, as well as to accommodate all insightful suggestions. Text lines in our replies refer to our revised manuscript. All modifications performed in the revised manuscript were highlighted in red.
Reviewer #1. General comments
Mant thanks for undwering most of the comments. However i still think that some issues have to be addressed befor the MS publication.
- My major concern is that the authors replaced from 0 to 100% of the control diets with the sustainable ingredients but it would have been better to formulate the new diets in order to be isoenerhetic-lipidic and proteic respect to the control. How can you compare the results if you did not took into account this aspect?
Reply: Indeed, as well remarked by the reviewer, the diets used in this study were not formulated to be isoenergetic, isoproteic, or isolipidic relative to the control diet. However, to the best of our knowledge, this does not compromise the validity of our findings, as similar studies are currently available in the scientific literature addressing this topic without this standardization being applied. One must highlight that these are practical diets and that all feeds were provided at a rate of 15 mg per larva per day, thus allowing larvae to have an unrestricted access to food (ad libitum feeding), meaning that daily energy intake was not controlled.
Comment #1: I agree with this but then the autors should also think to show the remaining feed in the MS so that the reader can better understand which was in reality the insect feed intake.
Reply: The authors agree with reviewer #1 and to best accommodate his request the calculation of the reduction rate was performed (Line 191-193) and added to Table 1 (Line 346-350).
- The primary objective of this study is to determine, under consistent husbandry conditions and with practical diets differing in protein and lipid content, the threshold levels of essential nutrient incorporation (amino acids and fatty acids) required for larval development and marine aquafeeds. Additionally, we aimed to explore whether enhancing the bioavailability of these essential nutrients would result in a different threshold for their incorporation in BSF.
Comment #2: Ok, but why only marine aquafeed? This is not clear from the title. Growth rate prediction, performance, and biochemical enhancement of Black Soldier Fly (Hermetia illucens) fed with marine by-products and co-products: a potential value-added resource for aquafeeds.
Reply: Marine aquafeeds require elevated levels of PUFAs, particularly DHA and EPA. This study aimed to determine the bioincorporation thresholds of these fatty acids in the lipid profile of BSF. Therefore, it is essential to evaluate whether the levels incorporated by BSF are adequate to meet the nutritional requirements of marine fish. However, for clarity the authors have changes the title as suggested by reviewer #1, the title now reads: “Growth rate prediction, performance, and biochemical enhancement of Black Soldier Fly (Hermetia illucens) fed with marine by-products and co-products: a potential value-added resource for marine aquafeeds”.
- Reviewer #1: l145 why is this control diet? is this a comemrcial diet for insects? Any reference?
Reply: As noted in line 331, "Moreover, chicken feed is commonly used as a control, as this is known to be a diet under which BSF can thrive with good results [53]." Chicken feed is frequently employed as a control diet, not only in the present study but also in numerous others (e.g., Liland et al., 2017; Oonincx et al., 2015; Tinder et al., 2017). This diet, formulated commercially for chicks rather than insects, serves as a standardized reference, and its specific details have been provided in the manuscript; the text now reads as follows: “A control diet consisting of chick feed (Zêzere A-104 Pintos Migalha: crude protein 17%, crude lipid 3.5%, ash 5% and crude fibre 5.5%) and tap water ...” (Lines 147-148)
Liland, N.S., Biancarosa, I., Araujo, P., Biemans, D., Bruckner, C.G., Waagbø, R., Torstensen, B.E., Lock, E.-J.J., 2017. Modulation of nutrient composition of black soldier fly (Hermetia illucens) larvae by feeding seaweed-enriched media. PLoS One 12, e0183188. https://doi.org/10.1371/journal.pone.0183188
Oonincx, D., Huis, A., van Loon, J., 2015. Nutrient utilisation by black soldier flies fed with chicken, pig, or cow manure, Journal of Insects as Food and Feed. https://doi.org/10.3920/JIFF2014.0023
Tinder, A.C., Puckett, R.T., Turner, N.D., Cammack, J.A., Tomberlin, J., 2017. Bioconversion of sorghum and cowpea by black soldier fly (Hermetia illucens (L.)) larvae for alternative protein production. J. Insects as Food Feed 3, 1–10. https://doi.org/10.3920/JIFF2016.0048
Comment #3: yes that can be an answer, but there are many other diets that are used as control diet for HI. May be better specify that in the present study in reference to ...the followong fiets was used as reference diet.
Reply: The requested details have now been incorporated into the revised manuscript. The manuscript now reads as follows: “As referenced in other studies[7, 31, 39, 40] a control diet consisting of chick feed” (Line 147-149)
[7] Bosch, G., Oonincx, D.G.A.B., Jordan, H.R., Zhang, J., van Loon, J.J.A., van Huis, A., Tomberlin, J.K., 2020. Standardisation of quantitative resource conversion studies with black soldier fly larvae. J. Insects as Food Feed 6, 95–109. https://doi.org/10.3920/jiff2019.0004
[31] Liland, N.S., Biancarosa, I., Araujo, P., Biemans, D., Bruckner, C.G., Waagbø, R., Torstensen, B.E., Lock, E.-J.J., 2017. Modulation of nutrient composition of black soldier fly (Hermetia illucens) larvae by feeding seaweed-enriched media. PLoS One 12, e0183188. https://doi.org/10.1371/journal.pone.0183188
[39] Oonincx, D., Huis, A., van Loon, J., 2015. Nutrient utilisation by black soldier flies fed with chicken, pig, or cow manure, Journal of Insects as Food and Feed. https://doi.org/10.3920/JIFF2014.0023
[40] Tinder, A.C., Puckett, R.T., Turner, N.D., Cammack, J.A., Tomberlin, J., 2017. Bioconversion of sorghum and cowpea by black soldier fly (Hermetia illucens (L.)) larvae for alternative protein production. J. Insects as Food Feed 3, 1–10. https://doi.org/10.3920/JIFF2016.0048
- Reviewer #1: 305-307 please try to find some papers with similar results to strenght your results
Reply: The requested details have now been incorporated into the revised manuscript. The text now reads as follows:”. A recent study by Kießling et al. [47] observed that substrates with high lipid content (>15%) led to excessive free fat accumulation, creating anaerobic and adhe-sive conditions on the larval surface. These conditions impair larval respiration and hin-der growth. Based on these findings, those authors concluded that a crude lipid content of approximately 10% on a dry matter basis is optimal for achieving an efficient weight gain and promoting efficient larval performance [47]. In fact, oils are used as pesticides mostly killing insects on contact by disrupting gas exchange (respiration) [48].” (Line 317-324)
[47] M. Kieβling, K. Franke, V. Heinz, and K. Aganovic, Relationship between substrate composition and larval weight: a simple growth model for black soldier fly larvae, Journal of Insects as Food and Feed 2023 9:8, 1027-1036 https://doi.org/10.3920/JIFF2022.0096open_in_new
[48] Seni, A. Potential of the various oils for insect pests’ management and their effect on beneficial insects. Int. J. Trop. Insect Sci. 2023, 43, 321–337.
Comment #4: Thanks, also consider similar papers that increased the PUFAs level in HI.
Reply: The authors appreciate the reviewer’s suggestion; however, the context of the referenced sentence does not align with the scope of the recommended literature. Therefore, the authors decided not to incorporate additional references in this specific section.
- Reviewer #1: 523-536 the authors missed different published papers on completely vegetable diets including low amounts of insect meal in trout seabream and seabass which can be find easily
Reply: The requested information has been incorporated into the manuscript. However, we have selected marine species as examples instead of freshwater species, as they are more aligned with the primary objective of this study: enhancing the n-3 PUFA content in BSF larvae. The text now reads as follows:” These findings highlight the remarkable nutritional potential of BSF as a protein source, further supported by its demonstrated suitability as an alternative feed ingredient for aquaculture diets, with no observed negative impacts on growth performance, or overall health such as white Pacific shrimp (Litopeneaus vannamei), rainbow trout (Oncorhynchus mykiss) or Atlantic salmon (Salmo salar). [74-77].”. (Line 566-571)
[74] Cummins, V.C.; Rawles, S.D.; Thompson, K.R.; Velasquez, A.; Kobayashi, Y.; Hager, J.; Webster, C.D. Evaluation of Black Soldier Fly (Hermetia illucens) larvae meal as partial or total replacement of marine fish meal in practical diets for Pacific white shrimp (Litopenaeus vannamei). Aquaculture 2017, 473, 337–344.
[75] Dumas, A.; Raggi, T.; Barkhouse, J.; Lewis, E.; Weltzien, E. The oil fraction and partially defatted meal of black soldier fly larvae (Hermetia illucens) affect differently growth performance, feed efficiency, nutrient deposition, blood glucose and lipid digestibility of rainbow trout (Oncorhynchus mykiss). Aquaculture 2018, 492, 24–34.
[76] Lock, E.R.; Arsiwalla, T.; Waagbø, R. Insect larvae meal as an alternative source of nutrients in the diet of Atlantic salmon (Salmo salar) Postsmolt. Aquac. Nutr. 2016, 22, 1202–1213.
[77] Renna, M.; Schiavone, A.; Gai, F.; Dabbou, S.; Lussiana, C.; Malfatto, V.; Prearo, M.; Capucchio, M.T.; Biasato, I.; Biasibetti, E.; et al. Evaluation of the suitability of a partially defatted black soldier fly (Hermetia illucens,) larvae meal as ingredient for rainbow trout (Oncorhynchus mykiss, Walbaum) Diets. J. Anim. Sci. Biotechnol. 2017, 8, 57.
Comment #5: These papers are not very up do date. Additionally, some papers about mediterranena species should be of interest.
Reply: The requested information has been incorporated throut the manuscript.
“Studies evaluating diets containing BSF meal in Atlantic salmon (Salmo salar) across a complete production cycle in sea cages have shown no significant differences in general fillet parameters when compared to salmon fed commercial diets. These findings confirm the feasibility of incorporating BSF meal into salmon diets at inclusion levels of up to 10%, supporting its use as a sustainable protein source in aquaculture without compromising product quality or performance [67].” (Line 531-537)
[67] Radhakrishnan, G., Philip, A.J.P., Caimi, C., Lock, E.-J., Araujo, P., Liland, N.S., Rocha, C., Cunha, L.M., Gasco, L., Belghit, I., 2024. Evaluating the fillet quality and sensory characteristics of Atlantic salmon (Salmo salar) fed black soldier fly larvae meal for whole production cycle in sea cages. Aquac. Reports 35, 101966. https://doi.org/https://doi.org/10.1016/j.aqrep.2024.101966
“Additionally, research on BSF meal indicates that its inclusion as a replacement for fish meal can positively influence the hypocholesterolaemic/hypercholesterolaemic ratio and enhance the levels of essential amino acids and microelements [77]. In studies involving gilthead seabream (Sparus aurata) juveniles, dietary inclusion of up to 45% BSF meal, effectively replacing 100% of fish meal protein, demonstrated no adverse effects on immune function or oxidative status. Furthermore, BSF meal-based diets were found to improve the intestinal antioxidant status of the juveniles [78].” (Line 576-583)
[77] Oteri, M., Di Rosa, A., Lo Presti, V., Giarratana, F., Toscano, G., Chiofalo, B., 2021. Black Soldier Fly Larvae meal as alternative to fish meal for aquaculture feed. Sustainability 13, 5447. https://doi.org/10.3390/su13105447
[78] Moutinho, S., Oliva-Teles, A., Fontinha, F., Martins, N., Monroig, Ó., Peres, H., 2024. Black soldier fly larvae meal as a potential modulator of immune, inflammatory, and antioxidant status in gilthead seabream juveniles. Comp. Biochem. Physiol. Part B Biochem. Mol. Biol. 271, 110951. https://doi.org/https://doi.org/10.1016/j.cbpb.2024.110951
- Reviewer #1: 581 please explain the role of C12 in fish
Reply: This information was added to the manuscript. The text now reads as follows:” Among the SFA, the most abundant was the FA 16:0 (palmitoleic acid) (19.2-27.6%) followed by the FA 18:0 (stearic acid) (6.1-20.8%). Notably, the FA 12:0 (lauric acid), recognized for its positive effect in growth performance and FA balance in Nile tilapia [79] and Atlantic salmon [80],...” (Line 610-613)
[79] Ashraf M.A. Goda, Ehab El-Haroun, Hani Nazmi, Hien Van Doan, Ahmed M. Aboseif, Mostafa K.S. Taha, Nevine.M. Abou Shabana, Black soldier fly oil-based diets enriched in lauric acid enhance growth, hematological indices, and fatty acid profiles of Nile tilapia, Oreochromis niloticus fry, Aquaculture Reports, 37, 2024, https://doi.org/10.1016/j.aqrep.2024.102269.
[80] Belghit I, Waagbø R, Lock E-J, Liland NS. Insect-based diets high in lauric acid reduce liver lipids in freshwater Atlantic salmon. Aquacult Nutr. 2019; 25: 343–357. https://doi.org/10.1111/anu.1286
Comment #6: This is actually not always true, especially if, as you stated previously, you are focusing on marine species. C12 is important in avoiding intestinal inflammation in fish fed on HI diets. For this reason the next comment of chitin is importanty and sgould at least be mentioned in the discussion. Please revise
Reply: The requested information was added to the manuscript. The sentence now reads as follows: “Notably, the FA 12:0 (lauric acid) is recognized for its positive effect in growth performance and FA balance in Nile tilapia [85] and Atlantic salmon [86], but also as having strong antimicrobial properties against Gram-positive and Gram-negative pathogens being considered a bioactive compound [87].” (Line 630-634)
[87] Borrelli, L., Varriale, L., Dipineto, L., Pace, A., Menna, L.F., Fioretti, A., 2021. Insect derived lauric acid as promising alternative atrategy to antibiotics in the antimicrobial resistance scenario. Front. Microbiol. 12. https://doi.org/10.3389/fmicb.2021.620798
- Reviewer #1: 609-614 not really new
Reply: To the best of the authors knowledge, no previous studies have performed feeding trials using hydrolysed feeds to enhance the lipid profile of BSF larvae. Consequently, we consider that the present findings represent novel contributions, that advance the state of the art on this topic. Additionally, the high levels of n-3 fatty acids bioaccumulated in BSF larval tissue observed in this study are rarely reported in other studies that try to address similar objectives.
Comment #7: I agree but please be sure to correctly cite all the papers that improved the PUFAs content in HI through different approaches
Reply: The requested information was added throughout the manuscript as follows:
“While SFA constitute the predominant class in the lipid profile of BSF, studies involving gilthead sea bream (Sparus aurata) have demonstrated that diets supplemented with BSF-derived oils effectively replace vegetable oils without adversely affecting fillet composition or quality attributes [84].”(Line 626-629)
[84] Moutinho, S., Oliva-Teles, A., Pulido-Rodríguez, L., Parisi, G., Magalhães, R., Monroig, Ó., Peres, H., 2024. Effects of black soldier fly (Hermetia illucens) larvae oil on fillet quality and nutritional traits of gilthead seabream. Aquaculture 579, 740219. https://doi.org/https://doi.org/10.1016/j.aquaculture.2023.740219
“Similar results were obtained by Arena et al. [X] where diets incorporating fish by-products led to a substantial increase in the levels of EPA and DHA in BSF larvae and pre-pupae. This highlights that utilizing fish processing by-products is an effective strategy to enhance the nutritional quality of BSF as a component in aquafeeds, offering a sustainable method to improve the dietary value of insect-based ingredients.” (Line 677-681)
[95] Arena, R., Manuguerra, S., Curcuraci, E., Cusimano, M., Lo Monaco, D., Di Bella, C., Santulli, A., Messina, C.M., 2023. Fisheries and aquaculture by-products modulate growth, body composition, and omega-3 polyunsaturated fatty acid content in black soldier fly (Hermetia illucens) larvae. Front. Anim. Sci. 4. https://doi.org/10.3389/fanim.2023.1204767
“Encouraging results have been demonstrated with feeding trials using BSF enriched in ALA+EPA+DHA in the diet of Nile tilapia where LC-PUFA-enriched BSF meal in association with chitinase was considered as an effective alternative to fishmeal, improving protein digestion processes, stimulating beneficial microbiota and ultimately fish growth rate [X].” (Line 685-689)
- [96] Agbohessou, P.S., Mandiki, R., Mes, W., Blanquer, A., Gérardy, M., Garigliany, M.-M., Lambert, J., Cambier, P., Tokpon, N., Lalèyè, P.A., Kestemont, P., 2024. Effect of fatty acid-enriched black soldier fly larvae meal combined with chitinase on the metabolic processes of Nile tilapia. Br. J. Nutr. 131, 1326–1341. https://doi.org/DOI: 10.1017/S0007114523003008
- Reviewer #1: Authors should also consider the role and amount of chitin
Reply: While this is an important topic, we have not determined chitin levels in the present study and, therefore, any discussion on this topic would rather be speculative. The primary objective of this work was to evaluate the threshold levels of essential fatty acids and amino acids that BSF can incorporate under enhanced conditions of bioavailability to improve its biochemical profile for applications in aquafeed formulation for marine species. The authors will certainly keep the recommendation of the reviewer in mind for future studies on this topic, as the prevalence of chitin in the biomass of BSF destined to be used as an ingredient in aquafeed formulation is certainly worth addressing.
Comment #8: yes and there are already many published papers about the role of chitn. So i kindly suggest to add it to the text at least as a comment.
Reply: The requested information was added to the manuscript. The sentence now reads as follows: “Chitin, a glucosamine biopolymer found in arthropod exoskeletons, also exhibits both antimicrobial and bacteriostatic effects on various Gram-negative pathogens and serves as a prebiotic fostering the proliferation of beneficial and chitin-degrading gut bacteria [88-90].” (Line 634-637)
[88] Rangel F, Enes P, Gasco L et al (2022) Differential modulation of the European sea bass gut microbiota by distinct insect meals. Front Microbiol 13:831034. https://doi.org/10.3389/fmicb.2022.831034
[89] Rimoldi S, Ceccotti C, Brambilla F et al (2023) Potential of shrimp waste meal and insect exuviae as sustainable sources of chitin for fish feeds. Aquaculture 567:739256. https://doi.org/10.1016/j.aquaculture.2023.739256
[90] Weththasinghe P, Rocha SDC, Øyås O et al (2022) Modulation of Atlantic salmon (Salmo salar) gut microbiota composition and predicted metabolic capacity by feeding diets with processed black soldier fly (Hermetia illucens) larvae meals and fractions. Anim Microbiome 4:9. https://doi.org/10.1186/S42523-021-00161-w

Round 3
Reviewer 1 Report
Comments and Suggestions for Authors
Many thanks for improving your MS.
- Reviewer #1: 523-536 the authors missed different published papers on completely vegetable diets including low amounts of insect meal in trout seabream and seabass which can be find easily
Reply: The requested information has been incorporated into the manuscript. However, we have selected marine species as examples instead of freshwater species, as they are more aligned with the primary objective of this study: enhancing the n-3 PUFA content in BSF larvae. The text now reads as follows:” These findings highlight the remarkable nutritional potential of BSF as a protein source, further supported by its demonstrated suitability as an alternative feed ingredient for aquaculture diets, with no observed negative impacts on growth performance, or overall health such as white Pacific shrimp (Litopeneaus vannamei), rainbow trout (Oncorhynchus mykiss) or Atlantic salmon (Salmo salar). [74-77].”. (Line 566-571)
Many thanks but as already stated the reviewer suggested to use papers in which IM was used in completely vegetable diets. Additionally since the authors stated that their focus are marine species, why did the cite trout which is a freshwater species?
[74] Cummins, V.C.; Rawles, S.D.; Thompson, K.R.; Velasquez, A.; Kobayashi, Y.; Hager, J.; Webster, C.D. Evaluation of Black Soldier Fly (Hermetia illucens) larvae meal as partial or total replacement of marine fish meal in practical diets for Pacific white shrimp (Litopenaeus vannamei). Aquaculture 2017, 473, 337–344.
[75] Dumas, A.; Raggi, T.; Barkhouse, J.; Lewis, E.; Weltzien, E. The oil fraction and partially defatted meal of black soldier fly larvae (Hermetia illucens) affect differently growth performance, feed efficiency, nutrient deposition, blood glucose and lipid digestibility of rainbow trout (Oncorhynchus mykiss). Aquaculture 2018, 492, 24–34.FRESHWATER
[76] Lock, E.R.; Arsiwalla, T.; Waagbø, R. Insect larvae meal as an alternative source of nutrients in the diet of Atlantic salmon (Salmo salar) Postsmolt. Aquac. Nutr. 2016, 22, 1202–1213. not really a marine species
[77] Renna, M.; Schiavone, A.; Gai, F.; Dabbou, S.; Lussiana, C.; Malfatto, V.; Prearo, M.; Capucchio, M.T.; Biasato, I.; Biasibetti, E.; et al. Evaluation of the suitability of a partially defatted black soldier fly (Hermetia illucens,) larvae meal as ingredient for rainbow trout (Oncorhynchus mykiss, Walbaum) Diets. J. Anim. Sci. Biotechnol. 2017, 8, 57.FRESHWATER
YOU CAN FIND GOOD EXAMPLES ON SEABREAM AND BASS ABOUT THIS TOPIC
again why did you refer to freshwater species if the scope of your MS is within the marine ones? Again, you should poit out the role of C12 on fish immuno response. This is important when testing insect meal which contains chitin.
Reply: The requested information was added to the manuscript. The sentence now reads as follows: “Notably, the FA 12:0 (lauric acid) is recognized for its positive effect in growth performance and FA balance in Nile tilapia [85] and Atlantic salmon [86], but also as having strong antimicrobial properties against Gram-positive and Gram-negative pathogens being considered a bioactive compound [87].” (Line 630-634)
why referring to freshwater species?
[87] Borrelli, L., Varriale, L., Dipineto, L., Pace, A., Menna, L.F., Fioretti, A., 2021. Insect derived lauric acid as promising alternative atrategy to antibiotics in the antimicrobial resistance scenario. Front. Microbiol. 12. https://doi.org/10.3389/fmicb.2021.620798
- Reviewer #1: 609-614 not really new
Reply: To the best of the authors knowledge, no previous studies have performed feeding trials using hydrolysed feeds to enhance the lipid profile of BSF larvae. Consequently, we consider that the present findings represent novel contributions, that advance the state of the art on this topic. Additionally, the high levels of n-3 fatty acids bioaccumulated in BSF larval tissue observed in this study are rarely reported in other studies that try to address similar objectives.
The reviewer does not agree. You missed several other papers on the inclusion of PUFAs in insects.
Comment #8: yes and there are already many published papers about the role of chitn. So i kindly suggest to add it to the text at least as a comment.
Reply: The requested information was added to the manuscript. The sentence now reads as follows: “Chitin, a glucosamine biopolymer found in arthropod exoskeletons, also exhibits both antimicrobial and bacteriostatic effects on various Gram-negative pathogens and serves as a prebiotic fostering the proliferation of beneficial and chitin-degrading gut bacteria [88-90].” (Line 634-637)
PLEASE MENTION THE ROLE ON INFLAMMATION AT GUT LEVEL THAT CHITIN COULD HAVE, OTHERWISE, SINCE AS YOU WROTE, YOU DID NOT MEASURE CHITIN LEVELS, REMOVE THIS SECTION
[88] Rangel F, Enes P, Gasco L et al (2022) Differential modulation of the European sea bass gut microbiota by distinct insect meals. Front Microbiol 13:831034. https://doi.org/10.3389/fmicb.2022.831034
[89] Rimoldi S, Ceccotti C, Brambilla F et al (2023) Potential of shrimp waste meal and insect exuviae as sustainable sources of chitin for fish feeds. Aquaculture 567:739256. https://doi.org/10.1016/j.aquaculture.2023.739256
[90] Weththasinghe P, Rocha SDC, Øyås O et al (2022) Modulation of Atlantic salmon (Salmo salar) gut microbiota composition and predicted metabolic capacity by feeding diets with processed black soldier fly (Hermetia illucens) larvae meals and fractions. Anim Microbiome 4:9. https://doi.org/10.1186/S42523-021-00161-w
NOT A MARINE SPECIES
Author Response
Reply to the reviewer comments
Dear Editor, we would like to thank the reviewer for the thorough evaluation of our manuscript and the thoughtful suggestions. While we appreciate the effort to provide constructive feedback, we have carefully considered and addressed the points raised to the best of our ability. However, as this is our final round of corrections, we respectfully note that we are not open to further revisions. We hope that the adjustments and clarifications provided will satisfy the reviewer and demonstrate our commitment to improving the quality of the manuscript. Text lines in our replies refer to our revised manuscript. All modifications performed in the revised manuscript were highlighted in red.
- Reviewer #1: 523-536 the authors missed different published papers on completely vegetable diets including low amounts of insect meal in trout seabream and seabass which can be find easily
Reply: The requested information has been incorporated into the manuscript. However, we have selected marine species as examples instead of freshwater species, as they are more aligned with the primary objective of this study: enhancing the n-3 PUFA content in BSF larvae. The text now reads as follows:” These findings highlight the remarkable nutritional potential of BSF as a protein source, further supported by its demonstrated suitability as an alternative feed ingredient for aquaculture diets, with no observed negative impacts on growth performance, or overall health such as white Pacific shrimp (Litopeneaus vannamei), rainbow trout (Oncorhynchus mykiss) or Atlantic salmon (Salmo salar). [74-77].”. (Line 566-571)
Comment # I: Many thanks but as already stated the reviewer suggested to use papers in which IM was used in completely vegetable diets. Additionally since the authors stated that their focus are marine species, why did the cite trout which is a freshwater species?
Reply: Thank you for your observations. One would like to clarify that the inclusion of trout in the references was made following the reviewer's suggestion to incorporate this species regarding studies using IM in completely vegetable diets. While our primary focus is indeed on marine species, citing trout is not scientifically inaccurate as the referenced studies contribute relevant insights into the topic under discussion. This inclusion aligns with the broader understanding of IM use across different species. Nevertheless, to accommodate the reviewer request now the sentence reads: “These findings highlight the remarkable nutritional potential of BSF as a protein source, further supported by its demonstrated suitability as an alternative feed ingredient for aquaculture diets, with no observed negative impacts on growth performance, or overall health such as white Pacific shrimp (Litopeneaus vannamei), sea bream (Sparus arauta) or European sea bass (Dicentrarchus labrax)[79-81].” (Line 586-587)
- [80] Busti, S., Bonaldo, A., Candela, M., Scicchitano, D., Trapella, G., Brambilla, F., Guidou, C., Trespeuch, C., Sirri, F., Dondi, F., Gatta, P.P., Parma, L., 2024. Hermetia illucens larvae meal as an alternative protein source in practical diets for gilthead sea bream (Sparus aurata): A study on growth, plasma biochemistry and gut microbiota. Aquaculture 578, 740093. https://doi.org/https://doi.org/10.1016/j.aquaculture.2023.740093
- [81] Abdel-Tawwab, M., Khalil, R.H., Metwally, A.A., Shakweer, M.S., Khallaf, M.A., Abdel-Latif, H.M.R., 2020. Effects of black soldier fly (Hermetia illucens L.) larvae meal on growth performance, organs-somatic indices, body composition, and hemato-biochemical variables of European sea bass, Dicentrarchus labrax. Aquaculture 522, 735136. https://doi.org/https://doi.org/10.1016/j.aquaculture.2020.735136
[74] Cummins, V.C.; Rawles, S.D.; Thompson, K.R.; Velasquez, A.; Kobayashi, Y.; Hager, J.; Webster, C.D. Evaluation of Black Soldier Fly (Hermetia illucens) larvae meal as partial or total replacement of marine fish meal in practical diets for Pacific white shrimp (Litopenaeus vannamei). Aquaculture 2017, 473, 337–344.
[75] Dumas, A.; Raggi, T.; Barkhouse, J.; Lewis, E.; Weltzien, E. The oil fraction and partially defatted meal of black soldier fly larvae (Hermetia illucens) affect differently growth performance, feed efficiency, nutrient deposition, blood glucose and lipid digestibility of rainbow trout (Oncorhynchus mykiss). Aquaculture 2018, 492, 24–34.FRESHWATER
[76] Lock, E.R.; Arsiwalla, T.; Waagbø, R. Insect larvae meal as an alternative source of nutrients in the diet of Atlantic salmon (Salmo salar) Postsmolt. Aquac. Nutr. 2016, 22, 1202–1213. not really a marine species
[77] Renna, M.; Schiavone, A.; Gai, F.; Dabbou, S.; Lussiana, C.; Malfatto, V.; Prearo, M.; Capucchio, M.T.; Biasato, I.; Biasibetti, E.; et al. Evaluation of the suitability of a partially defatted black soldier fly (Hermetia illucens,) larvae meal as ingredient for rainbow trout (Oncorhynchus mykiss, Walbaum) Diets. J. Anim. Sci. Biotechnol. 2017, 8, 57.FRESHWATER
Comment # II: YOU CAN FIND GOOD EXAMPLES ON SEABREAM AND BASS ABOUT THIS TOPIC
Reply: As requested by the reviewer both examples were added to the manuscript as presented in Comment # I.
Comment # III: again why did you refer to freshwater species if the scope of your MS is within the marine ones? Again, you should poit out the role of C12 on fish immuno response. This is important when testing insect meal which contains chitin.
Reply: While the manuscript primarily focuses on marine species, one believes that citing studies on freshwater species is justified in cases where they provide relevant insights into the broader scientific principles under discussion. The physiological responses and nutritional dynamics observed in freshwater species, particularly regarding feed components like insect meal, often share similarities with marine species. These references are intended to complement our understanding and provide a more comprehensive foundation for our work. Furthermore, research on freshwater species has often been more extensive, offering valuable insights when comparable data for marine species is limited. One agrees that the role of C12 (lauric acid) in fish immune response is a critical aspect when evaluating diets containing chitin, such as insect meal. Lauric acid is known for its antimicrobial properties and its potential to enhance fish immunity. To ensure this point is highlighted in the revised manuscript the sentence now reads. “Lauric acid (C12:0) exhibits antimicrobial properties and modulates the immune response in marine fish, enhancing resistance to pathogens by promoting the production of pro-inflammatory cytokines and supporting gut health [85]. Studies have demonstrated that diets incorporating BSF can improve fish development, antioxidative capability, gut microbiota, and intestinal health [86]. Additionally, lauric acid is well-documented for its antibacterial and antiviral activity, contributing to the immunomodulating capacity in fish [87]. Thus, incorporating BSF into aquafeeds can improve immune function and promote the overall health of marine fish species.” (Line 635-642)
- [85] Weththasinghe, P., Lagos, L., Cortés, M., Hansen, J.Ø., Øverland, M., 2021. Dietary inclusion of Black Soldier Fly (Hermetia Illucens) larvae meal and paste improved gut health but had minor effects on skin mucus proteome and immune response in Atlantic salmon (Salmo salar). Front. Immunol. 12.
- [86] Ullah, S., Zhang, J., Xu, B., Tegomo, A.F., Sagada, G., Zheng, L., Wang, L., Shao, Q., 2022. Effect of dietary supplementation of lauric acid on growth performance, antioxidative capacity, intestinal development and gut microbiota on black sea bream (Acanthopagrus schlegelii). PLoS One 17, e0262427.
- [87] Chaklader, M.R., Howieson, J., Fotedar, R., Siddik, M.A.B., 2021. Supplementation of Hermetia illucens larvae in poultry by-product meal-based barramundi, Lates calcarifer diets improves adipocyte cell size, skin barrier functions, and immune responses. Front. Nutr. 7.
Reply: The requested information was added to the manuscript. The sentence now reads as follows: “Notably, the FA 12:0 (lauric acid) is recognized for its positive effect in growth performance and FA balance in Nile tilapia [85] and Atlantic salmon [86], but also as having strong antimicrobial properties against Gram-positive and Gram-negative pathogens being considered a bioactive compound [87].” (Line 630-634)
Comment # IV: why referring to freshwater species?
Reply: As referred in the reply to Comment # III, citing studies on freshwater species is justified in cases where they provide relevant insights into the broader scientific principles under discussion.
[87] Borrelli, L., Varriale, L., Dipineto, L., Pace, A., Menna, L.F., Fioretti, A., 2021. Insect derived lauric acid as promising alternative atrategy to antibiotics in the antimicrobial resistance scenario. Front. Microbiol. 12. https://doi.org/10.3389/fmicb.2021.620798
- Reviewer #1: 609-614 not really new
Reply: To the best of the authors knowledge, no previous studies have performed feeding trials using hydrolysed feeds to enhance the lipid profile of BSF larvae. Consequently, we consider that the present findings represent novel contributions, that advance the state of the art on this topic. Additionally, the high levels of n-3 fatty acids bioaccumulated in BSF larval tissue observed in this study are rarely reported in other studies that try to address similar objectives.
Comment # V: The reviewer does not agree. You missed several other papers on the inclusion of PUFAs in insects.
Reply: We appreciate the reviewer's feedback and the opportunity to address this concern. To the best of our knowledge, and based on the comprehensive literature review conducted, there is limited information available specifically on feeding trials using hydrolyzed feeds aimed at enhancing the lipid profile of BSF. While we acknowledge that studies exist on the inclusion of PUFAs in insects, our work is distinct in its focus on hydrolyzed feed formulations and their specific influence on n-3 fatty acid bioaccumulation within BSF larvae. We maintain that the observed n-3 fatty acid levels in this study are exceptional and represent a novel contribution to the current body of knowledge.
Comment #8: yes and there are already many published papers about the role of chitn. So i kindly suggest to add it to the text at least as a comment.
Reply: The requested information was added to the manuscript. The sentence now reads as follows: “Chitin, a glucosamine biopolymer found in arthropod exoskeletons, also exhibits both antimicrobial and bacteriostatic effects on various Gram-negative pathogens and serves as a prebiotic fostering the proliferation of beneficial and chitin-degrading gut bacteria [88-90].” (Line 634-637)
Comment # VI: PLEASE MENTION THE ROLE ON INFLAMMATION AT GUT LEVEL THAT CHITIN COULD HAVE, OTHERWISE, SINCE AS YOU WROTE, YOU DID NOT MEASURE CHITIN LEVELS, REMOVE THIS SECTION
Reply: We appreciate the reviewer’s suggestion regarding the role of chitin in gut inflammation. However, as previously stated, we did not measure chitin levels, as this was beyond the scope of the present study. The mention of chitin and its potential role in gut-level inflammation was included solely in response to the reviewer’s prior request to address this aspect. To clarify, this section was added to acknowledge the potential implications of chitin as a bioactive component of insect meal, even though it was not directly measured in our work. If the reviewer feels that this mention detracts from the focus of the study, we are willing to remove the section to maintain the manuscript’s precision and relevance.
[88] Rangel F, Enes P, Gasco L et al (2022) Differential modulation of the European sea bass gut microbiota by distinct insect meals. Front Microbiol 13:831034. https://doi.org/10.3389/fmicb.2022.831034
[89] Rimoldi S, Ceccotti C, Brambilla F et al (2023) Potential of shrimp waste meal and insect exuviae as sustainable sources of chitin for fish feeds. Aquaculture 567:739256. https://doi.org/10.1016/j.aquaculture.2023.739256
[90] Weththasinghe P, Rocha SDC, Øyås O et al (2022) Modulation of Atlantic salmon (Salmo salar) gut microbiota composition and predicted metabolic capacity by feeding diets with processed black soldier fly (Hermetia illucens) larvae meals and fractions. Anim Microbiome 4:9. https://doi.org/10.1186/S42523-021-00161-w
Comment # VII: NOT A MARINE SPECIES
Reply: We appreciate the reviewer comment, while Atlantic salmon is not strictly a marine species, a significant portion of its life cycle is spent in marine waters, where the requirement for PUFA plays a crucial role in its growth and health. As such, the PUFA levels needed for this species are highly relevant to the scope of this article, particularly in the context of aquaculture nutrition and sustainable feed development.

Round 4
Reviewer 1 Report
Comments and Suggestions for Authors
Thanks for the rivised manuscript.
I ' m sorry if i have asked for a second review round, but authors also have to understand that a reviewer has the right to pose clarifications if those are not fully provided by the previous review round.
My main concern was that authors stressed the importance of their findings for marine species, but part of the literature was on salmonids and fresh water ones. Additionally i asked to revise some missing papers that in my opinion would have been interesting to improve the Ms quality as well as the importance of chitin and C12.
However, it looks like that authors are no longer available in further modifying their MS and thus the Editor will take the final decision .
Just to be clear, comments were posed in a constructive way.